# Temperature heterogeneity correlates with intraspecific variation in physiological flexibility in a small endotherm

Maria Stager [1,2 ✉], Nathan R. Senner [2], David L. Swanson[3], Matthew D. Carling[4], Douglas K. Eddy[4], Timothy J. Greives[5] & Zachary A. Cheviron[1]

Phenotypic flexibility allows individuals to reversibly modify trait values and theory predicts an individual's relative degree of flexibility positively correlates with the environmental heterogeneity it experiences. We test this prediction by integrating surveys of population genetic and physiological variation with thermal acclimation experiments and indices of environmental heterogeneity in the Dark-eyed Junco (*Junco hyemalis*) and its congeners. We combine field measures of thermogenic capacity for 335 individuals, 22,006 single nucleotide polymorphisms genotyped in 181 individuals, and laboratory acclimations replicated on five populations. We show that *Junco* populations: (1) differ in their thermogenic responses to temperature variation in the field; (2) harbor allelic variation that also correlates with temperature heterogeneity; and (3) exhibit intra-specific variation in thermogenic flexibility in the laboratory that correlates with the heterogeneity of their native thermal environment. These results provide comprehensive support that phenotypic flexibility corresponds with environmental heterogeneity and highlight its importance for coping with environmental change.

[1] Division of Biological Sciences, University of Montana, Missoula, MT, USA. [2] Department of Biological Sciences, University of South Carolina, Columbia, SC, USA. [3] Department of Biology, University of South Dakota, Vermillion, SD, USA. [4] Department of Zoology and Physiology, University of Wyoming, Laramie, WY, USA. [5] Department of Biological Sciences, North Dakota State University, Fargo, ND, USA. ✉email: mstager@mailbox.sc.edu

Temporal environmental variation imposes changing selection pressures that can result in multiple fitness optima occurring over time. In an effort to meet these changing fitness optima, an individual may express multiple trait values in the form of phenotypic flexibility (i.e., the ability to reversibly modify a trait value)[1]. Phenotypic flexibility provides an individual with repeated opportunities to match its phenotype to environmental conditions across its lifetime, and is predicted to evolve in environments characterized by frequent and predictable variation[2,3]. Accordingly, flexibility can represent an adaptive acclimatization response to environmental variation, especially for long-lived organisms[4]. Determining the causes and consequences of variation in flexibility among individuals is therefore crucial to our understanding of evolutionary processes and our ability to predict species' resilience to environmental change.

Because flexibility should increase fitness in variable environments, theory suggests that the magnitude of flexibility that individuals exhibit should positively correlate with the amount of environmental heterogeneity they experience[5–7]. If flexibility can be locally adapted in this way, empirical evidence for this theory should support three predictions: (1) geographic variation in flexibility corresponds with environmental heterogeneity, (2) genetic variation underlies this phenotypic variation, such that differences in flexibility can be replicated under common garden conditions, and (3) the degree of flexibility should be subjected to natural selection due to its influence on fitness, resulting in a reduction of variability in flexibility for populations occupying more heterogenous environments. Although there is some evidence that geographic variation in the degree of flexibility is associated with spatial patterns of environmental heterogeneity[8–10] (i.e., Prediction 1), there are very few empirical demonstrations supporting the other two predictions.

One common axis along which many environments vary is temperature. Whether daily or seasonally, nearly all environments vary in temperature over time. To meet the demands imposed by ambient thermal variation, most organisms exhibit behavioral and physiological responses to maintain homeostasis. For instance, many ectotherms alter their activity patterns and use of microclimates in response to changes in their thermal environment[11–13]. While endotherms can make similar behavioral alterations, they can also flexibly adjust their physiological ability to endogenously produce heat[14,15]. The ability to endogenously regulate body temperature can buffer endotherms from ambient thermal variation but it also comes with substantial costs.

Temperate environments, in particular, place a premium on endogenous heat production in small homeothermic endotherms[16]. Small birds and mammals commonly increase their thermogenic capacity (the ability to generate heat) during high-latitude winters. Failure to achieve adequate thermogenic output can have dire consequences for organismal fitness[17–19]. Increases in thermogenic capacity are also accompanied by physiological changes occurring at lower levels of biological organization[20–23]. Thus, a higher thermogenic capacity may be energetically costly to maintain due to the additional metabolic machinery required to achieve elevated thermogenesis[24]. Phenotypic flexibility in thermogenic capacity could therefore help mediate a balance between thermoregulation and its associated maintenance costs in response to fluctuating selective pressures[25]. Despite these expectations, it is unknown whether an individual's capacity for thermogenic flexibility is influenced by the degree of thermal variability it experiences throughout the year.

To test the influence of thermal heterogeneity on variation in thermogenic flexibility, we leveraged the natural environmental variation present across the dark-eyed Junco (*Junco hyemalis*) distribution. *J. hyemalis* is a temperate songbird comprised of

14 subspecies that inhabit a broad range of environments and exhibit conspicuous differences in life history, migratory tendency, physiology, size, song, plumage, and behavior[26–28]. Lab and field studies have demonstrated that *J. hyemalis* increases its thermogenic capacity in the cold[21,29,30] and this heightened thermogenic capacity is associated with a reduced risk of hypothermia[30,31]. We, therefore, expected that there may be variation in thermogenic flexibility among different *Junco* populations that correlates with the amount of temperature variation they experience.

In this study, we test this expectation by quantifying associations between environmental heterogeneity, population genetic structure, and flexibility. First, we present a survey of geographic variation in *Junco* thermogenic capacity (quantified as the maximum $O_2$ consumption under cold exposure, summit metabolism; $M_{sum}$) from field sites across the United States that host different *Junco* taxa. We include both *J. hyemalis* and *J. phaeonotus* because prior work indicates that the two clades are not reciprocally monophyletic[32]. We correlate these measures with recent weather data to determine which environmental indices correspond to variation in $M_{sum}$ in the field. If juncos exhibit geographic variation in $M_{sum}$ flexibility (Prediction 1), we expect to find a population × environment interaction in the field. We also expect that the magnitude of this flexibility (modeled as the slope of these population-level reactions norms[33]) will positively correlate with the annual thermal regime, which we use as a proxy for thermal heterogeneity. Second, we characterize fine-scale, range-wide population genetic variation within the *Junco* genus. We correlate variation in allele frequencies with environmental heterogeneity using genotype-environment association methods, which can aid in the detection of signatures of natural selection in instances where populations are not clearly distinguishable and environmental gradients are continuous[34]. We expect that, if there is genetic variation underlying potential differences in *Junco* flexibility (Prediction 2), genetic variation will be structured by the same environmental index that structures phenotypic variation. Finally, we present an acclimation experiment performed in the lab using five *Junco* populations to test whether the annual thermal regime (i.e., thermal heterogeneity) predicts the degree of flexibility they exhibit under controlled conditions while accounting for genetic relatedness among populations. We exposed individuals from each population to cold or control temperature treatments and quantified their thermogenic flexibility (defined as their change in $M_{sum}$ with acclimation). We expect that junco populations that experience greater seasonal temperature variation will exhibit higher $M_{sum}$ flexibility in the cold than those from more thermally stable regions (Prediction 2), represented by an interaction between temperature treatment and native temperature range. We also expect to find that populations from the most variable regions will show the least variability in their flexible response, resulting in lower coefficients of variation (CV) in $M_{sum}$ in the cold (Prediction 3). By combining these approaches, our results shed light on the ecological conditions that promote the evolution of increased flexibility and address long-standing hypotheses in the field of evolutionary biology.

## Results

**Intraspecific variation in thermogenic capacity**. To test whether juncos exhibit geographic variation in $M_{sum}$ flexibility, we assayed $M_{sum}$ for 335 juncos, including five *Junco* taxa (*J. h. caniceps*, *J. h. hyemalis*, *J. h. mearnsi*, *J. h. oreganus* group, and *J. p. palliatus*), at sites across the United States (Fig. 1a). The number of individuals, number of sampling days, seasons (breeding or non-breeding), and years varied across sites with 100 total site-days of

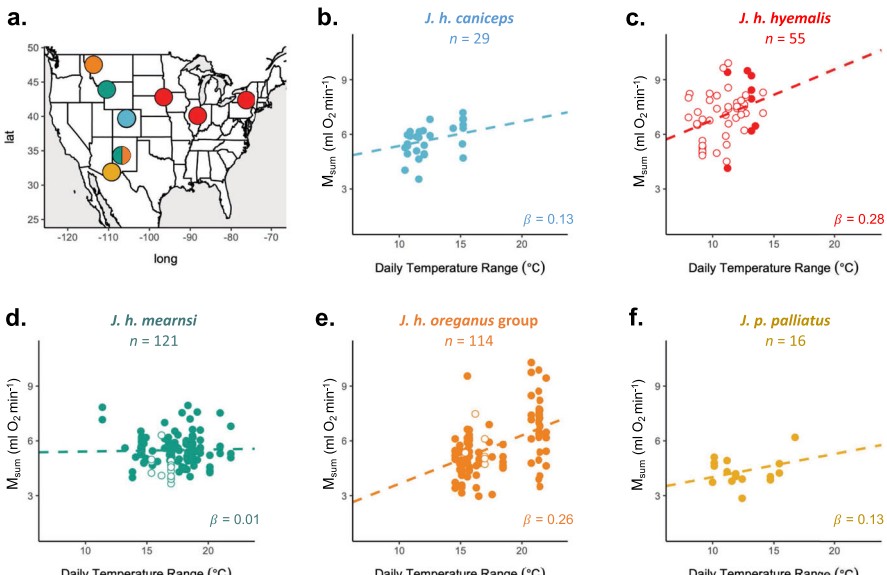

**Fig. 1 Relationships between daily temperature range and thermogenic capacity ($M_{sum}$) differ among five *Junco* taxa assayed in the field. a** Map of sampling sites for field measurements color-coded by taxon, following color scheme in **b–f** and plotted in R with package mapdata[87]. **b–f** Taxon-specific relationships between $M_{sum}$ and daily temperature range (averaged over the 8 days prior to capture), while controlling for differences in body mass. Each dot represents an individual ($n = 335$). Sample sizes for each taxon are listed at the top and the slope ($\beta$; in ml $O_2$ min$^{-1}$°C$^{-1}$) of each relationship is shown in the lower right corner. Sampling occurred during the breeding season (filled circle) or nonbreeding season (open circle). Lines fit from glm ($M_{sum} = M_b + Taxa \times T_{d\_range}$) using mean $M_b$ for each taxon. Source data are provided in the Source Data file.

environmental variation included in our dataset. Variation in $M_{sum}$ corresponded to recent weather variation, with mean daily temperature range ($T_{d\_range}$) in the 8 days prior to capture explaining more variation than the other environmental variables tested (Table S1). The most well-supported model, explaining 48% of the variation in $M_{sum}$ (AIC = 960.9), included body mass ($M_b$), $T_{d\_range}$, taxon, and $T_{d\_range} \times$ taxon. The addition of a term for the season did not improve model fit ($\Delta$AIC = 0.14; $\beta = -0.28 \pm 0.21$, $p = 0.18$). $M_b$ positively correlated with $M_{sum}$, such that larger individuals exhibited higher thermogenic capacity ($\beta = 1.17 \pm 0.15$ ml $O_2$ min$^{-1}$, $p = 1.1 \times 10^{-13}$). Furthermore, juncos that experienced greater $T_{d\_range}$ had elevated $M_{sum}$, suggesting that they may acclimatize to short-term heterogeneity in their thermal environment ($\beta = 1.87 \pm 0.24$, $p = 6.0 \times 10^{-14}$). The model also showed persistent differences in $M_{sum}$ among taxa while controlling for differences in $M_b$ and $T_{d\_range}$, with *J. p. pallitatus* exhibiting the lowest ($\beta = -1.36 \pm 0.48$, $p = 4.1 \times 10^{-3}$) and *J. h. hyemalis* the highest $M_{sum}$ values ($\beta = 2.23 \pm 0.46$, $p = 1.8 \times 10^{-6}$). Moreover, we also found that the inclusion of the $T_{d\_range} \times$ taxon interaction term substantially improved model fit indicating that some taxa respond to variation in $T_{d\_range}$ more strongly than others (model without interaction: $\Delta$AIC = 15.4; Fig. 1b–f). The magnitude of flexibility (i.e., the resulting slopes of the taxon-specific reaction norms from this best model) did not correlate with an index of thermal heterogeneity (measured as the mean annual temperature range across capture sites) among taxa ($R^2 = 0.10$, $p = 0.61$, $n = 5$). However, when *J. h. mearnsi* was removed from the analysis, we found a strong correlation between magnitude of flexibility and annual temperature range ($R^2 = 0.92$, $\beta = 0.02 \pm 0.004$, $p = 0.04$, $n = 4$; Fig. S1), as outlined in our first prediction.

**Genotype-environment associations**. To understand how population genetic variation across the genus might correspond to environmental variation, we genotyped 22,006 biallelic SNPs from 181 individuals representing all recognized *Junco* species

and subspecies (Fig. 2a). The first two axes from a principal component analysis (PCA) explained 7.5% of the total genetic variance. Major clusters identified by the PCA corresponded to the lineages of Costa Rica and Panama (*J. vulcani*), southern Mexico and Guatemala (*J. phaeonotus alticola* and *J. p. fulvescens*), southern Baja (*J. bairdi*), and Guadalupe Island (*J. insularis*), with all other taxa (*J. p. palliatus*, *J. p. phaeonotus*, and all *J. hyemalis*) grouping together (Fig. 2b, c). We subtracted the variance explained by PC1 and PC2 in our genotype-environment associations (performed using redundancy analysis; RDA) to control for population history and structure[35–37]. After controlling for background population structure, the remaining signatures of differentiation can be related to environmental variables and, as such, these relationships may reflect local adaptation to environmental conditions. In our partial RDA, eight axes composed of eight climatic variables explained 2.9% of the genetic variance across *Junco* (adj. $R^2$). Permutation tests confirmed the significance of the model ($F = 1.55$, df = 8, $p = 0.001$) and the first four RDA axes ($p \leq 0.05$; Fig. 2d, e). In support of our second prediction, a variance partition analysis of this model found that the annual temperature range explained more allelic variation than any other climatic variable tested (0.6%; Table 1).

**Common garden acclimation experiments**. To formally test whether phenotypic flexibility in $M_{sum}$ correlated with thermal heterogeneity, we performed a laboratory acclimation experiment using 95 individuals from five nonmigratory populations across the western United States (*J. h. aikeni*, *J. h. dorsalis*, *J. h. shufeldti*, *J. h. thurberi*, and *J. p. palliatus*). The sites at which these populations were captured varied by 21 °C in their annual temperature range ($T_{range}$; Fig. 3). Genetic differentiation among populations ($F_{ST}$) ranged from 0.019 to 0.051 (Fig. 3a), but pairwise environmental and genetic distances did not covary (partial Mantel test conditioned on geographic distance: $r = -0.40$, $p = 0.82$, $n = 5$). Prior to acclimation and following an

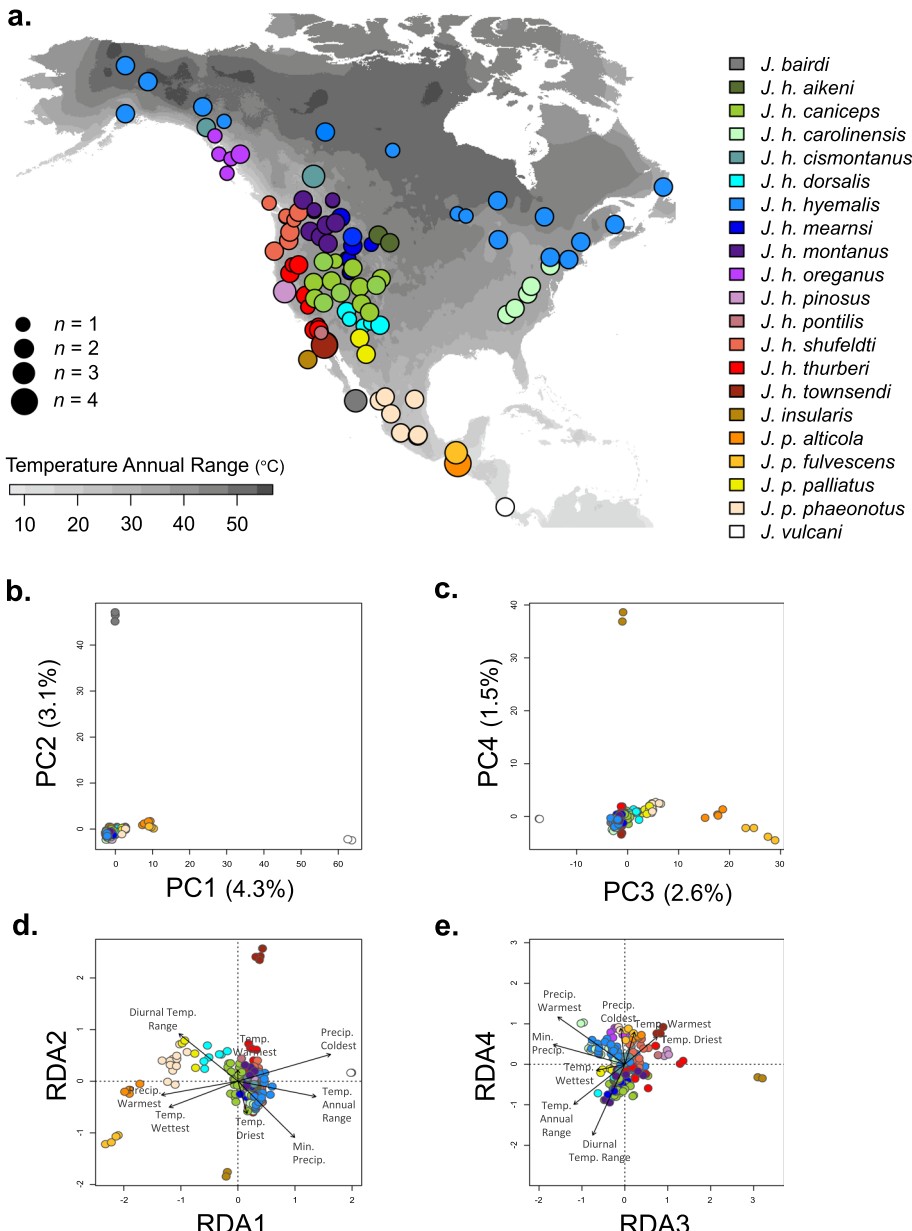

**Fig. 2 *Junco* population genetic variation. a** Origin of 181 *Junco* individuals included in the SNP dataset. Map shading (in grayscale) depicts temperature annual range values from WorldClim[66], plotted in R with package raster[67]. Dot color denotes museum-based taxon assignments; dot size represents the number of individuals from that locale ($n = 1$ to 4). Population genetic variation from 22,006 genome-wide SNPs structured along four PC axes (**b**, **c**) and four RDA axes (**d**, **e**) where each dot represents an individual. Colors follow the legend provided in (**a**). **d**, **e** Arrows indicate loadings of eight WorldClim variables (diurnal temperature range, temperature annual range, mean temperature of the warmest quarter, mean temperature of the wettest quarter, mean temperature of the driest quarter, mean minimum precipitation, mean precipitation of the warmest quarter, and mean precipitation of the coldest quarter) in a partial RDA conditioned on PC axes 1 and 2. Source data are provided in the Source Data file.

8-week adjustment period under common conditions at 23 °C, temperature treatment groups did not differ in $M_{sum}$ (posterior $\mu = 0.18$, 95% CI = $[-0.06, 0.40]$, $p = 0.14$) or $M_b$ (posterior $\mu = -0.07$, 95% CI = $[-0.57, 0.41]$, $p = 0.78$). However, both $M_{sum}$ and $M_b$ positively correlated with native $T_{range}$ ($M_{sum}$: posterior $\mu = 1.10$, 95% CI = $[0.26, 1.97]$, $p = 0.02$; $M_b$: posterior $\mu = 3.71$, 95% CI = $[3.07, 4.34]$, $p = 2.0 \times 10^{-4}$).

After a 3-week acclimation treatment to 3 °C (Cold) or 23 °C (Control), we found that $\Delta M_{sum}$ (measured as the difference between post- and pre-acclimation measures) was higher in larger and cold-acclimated birds ($M_b$: posterior $\mu = 0.53$, 95% CI = $[0.03, 1.00]$, $p = 0.03$; Cold Treatment: posterior $\mu = 0.62$, 95% CI

$= [0.35, 0.88]$, $p < 0.001$). In support of our second prediction, we also found an interaction between native $T_{range}$ and temperature treatment, such that populations from more variable thermal climates exhibited the greatest flexibility in $M_{sum}$ in the Cold (posterior $\mu = 0.60$, 95% CI = $[0.07, 1.14]$, $p = 0.03$), while populations from less variable thermal climates exhibited little or no change in $M_{sum}$ (Fig. 3b). In support of our third prediction, populations from less variable thermal environments also showed larger CVs in their flexibility in $M_{sum}$ in the cold than did those from more variable thermal environments ($R^2 = 0.80$, $\beta = -21.40 \pm 6.11$, $p = 0.04$, $n = 5$). This pattern was not evident in the Control group ($\beta = 0.34 \pm 0.18$, $p = 0.16$, $n = 5$).

**Table 1 Genetic variance across 22,006 genome-wide SNPs for 181 *Junco* individuals partitioned by eight WorldClim variables in a partial RDA (conditioned on background structure using the first two axes of a PCA).**

| WorldClim variable | Adj. $R^2$ |
|---|---|
| Diurnal temperature range | 0.0034 |
| Temperature annual range | 0.0057 |
| Temperature wettest quarter | 0.0008 |
| Temperature driest quarter | 0.0016 |
| Temperature warmest quarter | 0.0008 |
| Minimum precipitation | 0.0028 |
| Precipitation warmest quarter | 0.0029 |
| Precipitation coldest quarter | 0.0035 |
| **Constraining variable total** | **0.0294** |
| **Conditional variable total** | **0.0642** |

Total variance explained by the constraining (environmental) and conditional (PC1 and PC2) variables shown at the bottom.

## Discussion

Our multifaceted approach provides support for three key predictions of theory regarding local adaptation in phenotypic flexibility. We present evidence that *Junco* populations responded differentially to thermal cues in the field and that temperature variability may be driving these patterns of intraspecific variation in $M_{sum}$. We replicated this pattern in the laboratory where flexibility in *Junco* $M_{sum}$ correlated with the heterogeneity of their native thermal environment. Moreover, range-wide variation in allele frequencies also correlated with annual temperature range, providing evidence that thermal heterogeneity may be an important determinant of population structure across the *Junco* radiation. Together, these results greatly expand our knowledge of endothermic responses to environmental variation and their capacity for thermal acclimatization.

We first predicted that geographic variation in $M_{sum}$ flexibility would correspond with environmental heterogeneity. Several studies have characterized broad-scale interspecific patterns in endothermic $M_{sum}$[38–40], but far less is known about the potential for, or the underlying environmental correlates of, intraspecific variation in $M_{sum}$ (but see refs. [41–43]). We found that *Junco* taxa exhibited constitutive differences in $M_{sum}$ in the field even after controlling for variation in $T_{d\_range}$ and $M_b$ among taxa. We also observed variation among individuals in $M_{sum}$ in response to recent changes in ambient thermal conditions—including those occurring within a season. We would not expect this relationship between adult $M_{sum}$ and recent temperature range if the observed variation was attributable to interindividual differences alone. Thus, although we did not perform repeated measures on individuals to quantify flexibility directly, we believe that this pattern is indicative of a thermal acclimatization response. This result is consistent with recent laboratory findings showing that *J. h. montanus* can substantially increase $M_{sum}$ within 1 week of exposure to low temperatures[30]. Moreover, we found that these responses differ among *Junco* taxa, suggesting that *Junco* populations may differ in their physiological flexibility. Using the reaction norms from this relationship as a measure of the magnitude of flexibility, we found a positive correlation between thermogenic flexibility and annual thermal heterogeneity across four of the five taxa. Yet when the taxa with the lowest flexibility (*J. h. mearnsi*) is included, this correlation disappears. It is difficult to interpret these absolute differences among taxa because we were not able to control for several factors in the field (e.g., an individual's diet, reproductive status, use of microclimatic refugia, duration at the site prior to capture, etc.). Additionally, the measure of thermal heterogeneity that we applied is an imperfect

approximation of both operative temperature and what these birds likely experience throughout the year due to the migratory tendency of many individuals in this dataset. For example, *J. h. mearnsi* do not overwinter at the sites where we captured them during the breeding season (at elevations >2000 m near Teton National Park). However, we took this approach because we lacked data regarding the annual movements of each individual. Thus, to provide additional support for our result that temperature variability influenced differences in flexibility across the *Junco* distribution we also incorporated both genotype-environment association analyses and a laboratory acclimation experiment.

We next predicted that if the phenotypic patterns we observed were indeed a result of local adaptation to temperature heterogeneity, we would find genetic variation that was also structured by temperature range. Indeed, the annual temperature range at a sampling locality explained more genetic variation across the genus than any of the other climatic variables tested. Together, the eight climatic variables explained a small amount of total variation in allele frequencies. However, given that we only surveyed 2% of the genome with our SNP dataset, this amount is well within the range of variation expected from a genotype-environment association analysis (e.g., refs. [36,37,44]). Additionally, the climatic data that we employed corresponded to the annual climatic regimes at our collection sites, even though some individuals included in the analysis are likely migratory and therefore not present at these sites year-round. Once again, this approach was necessary because of uncertainty regarding where each individual spent the entire year. In the future, a more refined approach that accurately accounts for the annual environmental conditions experienced by each individual could therefore improve the explanatory power of this analysis.

We also predicted that differences in flexibility would be replicated under common garden conditions. Again, our ability to connect *Junco* populations to their native climatic regimes is restricted by our limited knowledge of their movements throughout the year; thus we chose to focus our acclimation study on populations that likely remain resident in one narrow geographic area, enabling us to reliably reconstruct their climatic histories. We selected these focal populations to maximize variation in the annual temperature range they experienced based on the results of our field sampling. Importantly, environmental and genetic variation among these populations did not covary, allowing us to simultaneously tease apart the effects of both factors. Furthermore, we accounted for nonindependence among populations by incorporating measures of population genetic differentiation in our acclimation analysis using Markov Chain Monte Carlo generalized linear mixed models[45,46]. The acclimation experiment, therefore, provided a powerful opportunity to test predictions of flexibility and environmental heterogeneity, and further permitted us to (1) perform repeated measures on individuals to formally quantify flexibility; (2) expose all individuals to the same temperature range stimulus; (3) control for acclimatization history for 2 months prior to temperature treatments; and (4) control for several other factors (e.g., diet, food availability, microclimatic refugia) that may influence observed flexibility. We found that individuals from the two coastal populations in southern California (*J. h. thurberi*) and Oregon (*J. h. shufeldti*), which experience the smallest amount of temperature variation, exhibited little to no flexibility in the cold. This inflexibility could result from a failure to sufficiently stress these populations, such that phenotypic change was not necessary to meet the challenge at hand. This does not appear to be the case though, as all individuals were hypothermic at the end of the $M_{sum}$ assay. The only population that was included in both the field and lab studies (*J. p. palliatus*), meanwhile, exhibited

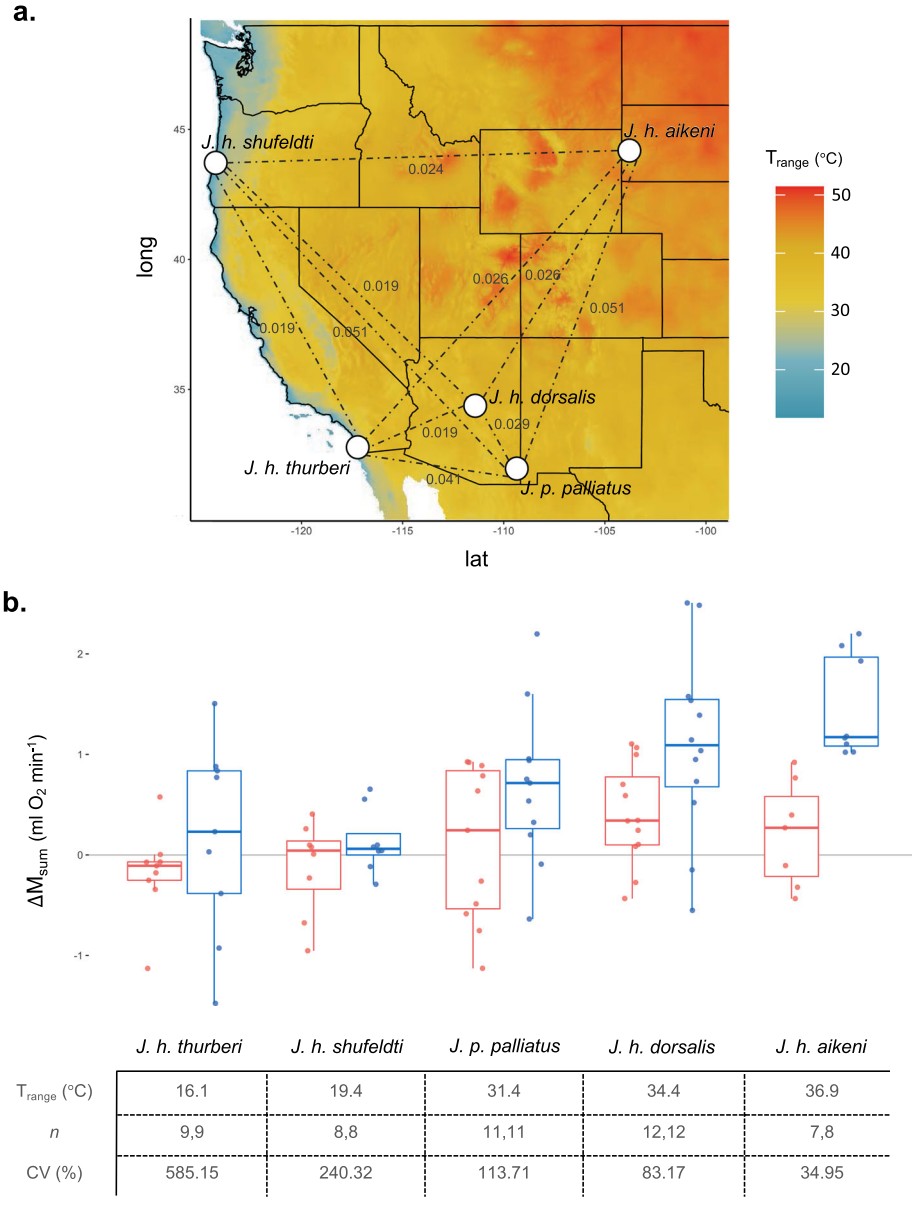

**Fig. 3 *Junco* thermogenic flexibility correlates with thermal heterogeneity. a** Origin of five *Junco* populations used in laboratory acclimations (dots) that differ in their native annual temperature range ($T_{range}$; values from WorldClim[66]), shown in shaded colors and plotted in R with package raster[67]. Pairwise $F_{ST}$ indicated by dotted lines and accompanying values in gray. **b** Thermogenic flexibility, measured as the change in $M_{sum}$ over the 3-week temperature acclimation treatment (post- minus pre-acclimation), for each population. Boxplots show the median values, the 25th and 75th percentiles (lower and upper margins of the box), and the minimum and maximum values ≤1.5 × IQR from the box margin (whiskers). Dots indicate individual measures ($n = 95$). Control birds (acclimated to 23 °C) in red; cold-acclimated birds (3 °C) in blue. For each population, the native $T_{range}$, number of individuals in each treatment (Control, Cold), and coefficient of variation for cold-acclimated individuals (CV) are shown below. Source data are provided in the Source Data file.

intermediate levels of flexibility in each instance. Finally, birds experiencing the greatest annual temperature variation—those from the Black Hills region (*J. h. aikeni*) followed by those from the Arizona highlands (*J. h. dorsalis*)—exhibited the greatest flexibility in their response to the cold. These results, therefore, provide strong support for the prediction that juncos from more thermally variable environments exhibit greater flexibility in $M_{sum}$.

Finally, we predicted that we would find a reduction in variability in flexibility for populations occupying more heterogenous environments. In agreement with this prediction, we found a strong negative correlation between the coefficient of variation for $M_{sum}$ flexibility and the annual temperature range.

Thus, the patterns observed here may be indicative of selection honing the flexible response in variable climates (i.e., lower CVs) and/or relaxed selection in invariable environments (i.e., higher CVs). Taken together, this work suggests that selection has shaped the junco's capacity to acclimatize to changes in ambient temperature. The ability to robustly respond to thermal challenges by achieving high $M_{sum}$ may impart survival advantages in the cold, and at the same time, the ability to downregulate metabolic capacity (i.e., $M_{sum}$) may provide energetic savings under conditions where high thermogenesis is not required.

Nonetheless, a necessary additional step in assessing whether or not flexibility can be locally adapted is to connect differences in

flexibility with direct measures of fitness. Unfortunately, we cannot directly address whether differences in flexibility in *Junco* $M_{sum}$ result in differential fitness outcomes with our dataset. Previous studies have shown that absolute measures of thermogenic capacity do positively correlate with survival rates in other temperate-wintering songbirds[17,18]. However, this is not analogous to linking thermogenic flexibility and fitness because it does not account for the potential costs of trait alteration (i.e., changing $M_{sum}$ over time). Large-scale demographic studies are therefore also needed to determine the strength of selection on thermogenic flexibility across different environments that vary in thermal heterogeneity.

Quantifying these direct connections between fitness and thermal acclimatization is vital in determining how endotherms can respond to changing environments[47]. For example, previous work has suggested that many ectotherms are limited in their thermal acclimatization capacity and this has been used as evidence that ectotherms are especially vulnerable to future climatic change[48–50]. There are several biological differences between ectotherms and endotherms that may contribute to potential disparities in their acclimatization ability, however. In general, many ectotherms rely on behavioral thermoregulatory mechanisms and possess a number of avoidance strategies (e.g., diapause, hibernation, or migration) that may be used to buffer against environmental extremes[51]. Thus, the amount of thermal heterogeneity that an individual or population experiences may not correspond to broad-scale climatic patterns across the year. Endotherms maintain a relatively constant body temperature in comparison, despite sometimes large temperature differentials with their ambient environment[16]. While many endotherms also exhibit hibernation and migratory behaviors, small songbirds that reside in temperate regions year-round, like juncos, are particularly exposed to seasonal temperature variation[25]. These differences could lead to divergent selection pressures on flexibility and thermal performance traits among taxonomic groups.

Ultimately, many of the models that are used to forecast the effects of global change on population dynamics[52], make habitat delineations[53], and model disease transmission[54] rely on accurate characterizations of variation in thermal tolerance. Several recent macrophysiological approaches characterizing potential organismal responses to climatic change employ a single metric of thermal tolerance, for instance, effectively treating tolerance as a trait that is canalized and invariant across a species' range and an individual's lifetime (e.g., refs. [50,55,56]). Our results instead highlight the capacity for populations to vary geographically in their physiological response to environmental cues. When coupled with datasets like ours, biophysical models that incorporate intraspecific patterns in acclimatization will improve our ability to predict organismal responses to climate change.

## Methods

### Field data and analysis
*Field sampling.* We captured adult juncos by mist net or potter trap at sites in Arizona (breeding season), Colorado (breeding), Illinois (non-breeding), Montana (breeding), New Mexico (non-breeding), New York (breeding), South Dakota (non-breeding), and Wyoming (breeding), spanning 16° in latitude and 37° in longitude (Fig. 1). This work was completed with approval from the U.S. Fish and Wildlife Service (MB84376B-1 to M.S.; MB01543B-0 to Z.A.C.; MB758442 to D.L.S.; MB06336A-4 to M.D.C.; MB757670-1 to David Winkler; and MB094297-0 to Christopher Witt,), the State of Arizona Game and Fish Department (SP590760 and SP707897 to D.L.S.), Colorado Parks and Wildlife (10TRb2030A15 to M.D.C.), the Illinois Department of Natural Resources (NH13.5667 to Z.A.C.), the Montana Department of Fish Wildlife and Parks (2016-013 and 2017-067-W to M.S.), the New Mexico Department of Game & Fish (3217 to Christopher Witt), the New York State Division of Fish, Wildlife, & Marine Resources (LCP 1477 to David Winkler), the State of South Dakota Department of Game, Fish, and Parks (06-03, 07-02, 08-03 to D.L.S.), Wyoming Game and Fish (754 to M.D.C.), and the Institutional Animal Care and Use Committees at Cornell University (2001-0051 to David Winkler), the University of Illinois (13385 to Z.A.C.), the University of

Montana (010-16ZCDBS-020916 to Z.A.C.), the University of South Dakota (03-08-06-08B to D.L.S.), and the University of Wyoming (A-3216-01 to M.D.C.). We classified individuals to taxonomic unit based on plumage (*J. h. caniceps, J. h. hyemalis, J. h. mearnsi, J. h. oreganus* group, and *J. p. palliatus*). The *J. h. oreganus* group encompasses seven subspecies (*J. h. montanus, J. h. oreganus, J. h. pinosus, J. h. pontilis, J. h. shufeldti, J. h. thurberi*, and *J. h. townsendii*) with similar plumages and overlapping nonbreeding ranges[26,27]. For this reason, we were unable to distinguish subspecies of nonbreeding individuals within this group, though all breeding individuals were collected from the *J. h. montanus* range.

*Field metabolic assays.* Birds were transported from the site of capture to a nearby laboratory (<50 mi away) for measurement where they were held indoors with ad libitum access to food and water. We assayed the $M_{sum}$ of each individual using open-flow respirometry. Most measurements were completed within 1 day of capture to avoid the effects of captivity on metabolic rates, though some ($n = 15$) were made within 2 days of capture (mean = 0.85 ± 0.47 day). $M_b$ (in g) was quantified before each measurement began. $M_{sum}$ trials were conducted during daylight hours. A single individual was placed in a metabolic chamber in a dark, temperature-controlled environment. We pumped dry, cooled heliox gas (21% $O_2$ and 79% He) through the animal's chamber at a constant flow rate. We subsampled the outflow current, dried it (Drierite), scrubbed it of $CO_2$ using ascarite, and dried it again before quantifying the $O_2$ concentration using a FoxBox (Sable Systems). We recorded the output using Expedata software v.1.9.13. Trials were conducted using static cold exposure (−5 °C) for Colorado, Illinois, Montana, New Mexico, New York, and Wyoming birds and sliding cold exposure (starting at 0 °C and descending 3 °C every 20 min) for Arizona and South Dakota birds; however, both methods have been shown to produce similar estimates of $M_{sum}$[57]. Trials lasted until $O_2$ consumption plateaued or declined for several minutes. We also sampled a blank chamber before and after trials to account for potential fluctuations in baseline, ambient air.

We used custom R scripts (https://github.com/Mstager/batch_processing_Expedata_files) to correct for drift in baseline $O_2$ content, calculate oxygen consumption ($VO_2$)[58], and quantify $M_{sum}$ (calculated as the highest instantaneous $VO_2$ averaged over a 5-min period; ml $O_2$ min$^{-1}$). We discarded measures characterized by large drift in baseline $O_2$ (owing to ambient temperature fluctuations affecting the FoxBox) or inconsistent flow rates resulting in a total sample size of $n = 335$ individuals. Following measurements, birds were subject to different fates: either released, exposed to acclimation experiments[30], or immediately euthanized and deposited in museums (Source Data). Metabolic data from South Dakota have been previously published[59].

*Environmental data for field sampling sites.* To account for an individual's recent acclimatization history, we retrieved weather data associated with each collection site (rounded to the nearest hundredth of a degree latitude/longitude) from the DayMet dataset using the R package daymetr v.1.4[60]. This dataset is composed of daily weather parameter estimates derived using interpolation and extrapolation from meteorological observations for a 1 km x 1 km gridded surface across North America[61]. We downloaded daily estimates of minimum temperature ($T_{min}$; °C), maximum temperature ($T_{max}$; °C), precipitation (prcp; mm/day), water vapor pressure (vp; Pa), shortwave radiation (srad; W/m$^2$), and daylength (dayl; s/day) for the 14 days prior to each individual's capture date. We additionally calculated daily temperature range ($T_{d\_range}$; °C) as $T_{max} - T_{min}$. We selected a conservative potential acclimatization window because, given their migratory nature, we do not know how long juncos were present at a site before sampling occurred, preventing us from incorporating more extensive windows (e.g., months). Prior work in *J. hyemalis* suggested a window of 7–14 days would be appropriate[30,62]. We, therefore, calculated averages across acclimatization windows varying from 7 to 14 days preceding capture (i.e., from 1–7 to 1–14) for each weather variable. We also retrieved elevation (elev; m) for each site using the package googleway v.2.7.3[63].

*Analyses for field data.* All analyses were conducted in R v.4.0.2[64]. To determine whether junco $M_{sum}$ varied with environmental variation, we constructed eight linear models with $M_b$, taxon, and a single environmental variable ($T_{min}$, $T_{max}$, prcp, dayl, vp, srad, or $T_{d\_range}$ averaged across the acclimatization window or elev) as main effects. We chose to use one variable at a time because several environmental variables were strongly correlated in our dataset (i.e., $T_{min}$ and $T_{max}$: $r = 0.96$). We first standardized continuous predictor variables by centering and dividing by two standard deviations using the package arm v.1.11-2[65]. As an indicator of differences in flexibility, we additionally included an interaction between the environmental variable and taxon to determine if taxa differed in their response to environmental cues. Thus models took the form $M_{sum} = M_b + \text{taxon} \times \text{environmental variable}$. We present the resulting standardized $\beta$ to characterize the effect sizes of these relationships. We then used AIC values to evaluate differences in model fit among environmental variables and with that of a null model (including only $M_b$ and taxon as predictors) in order to identify a single "best" environmental variable and acclimatization window explaining variation in $M_{sum}$. To justify the inclusion of the interaction term, we compared models with and without the term using ΔAIC. We also used ΔAIC to test the inclusion of a variable for the season in which we sampled each

individual (breeding or nonbreeding). The *J. h. oreganus* group served as the reference taxa in reported values.

Lastly, following the results of those analyses, which emphasized the importance of thermal variability, we tested whether junco thermogenic flexibility in the field corresponded with the heterogeneity of their thermal environment. As an index of thermal heterogeneity, for each individual, we downloaded the annual temperature range (˚C) for the site of capture from the WorldClim database[66], composed of spatially interpolated climate data aggregated from 1970–2000, at a resolution of 2.5′ using the R package raster v.3.3-13[67]. Using these values, we calculated the mean annual temperature range for each taxon. We then correlated the reaction norms resulting from our best model (i.e., the taxon-specific slopes; ml $O_2$/min/˚C) with the mean annual temperature range for each taxon using a linear regression.

### Population genetic data

*Sampling, sequencing, and SNP generation.* For phylogeographic reconstruction, we obtained muscle tissue samples ($n = 192$) from museum specimens collected across the breeding distribution of all 21 *Junco* taxa identified by Miller[26]. We tried to maximize the environmental variance in our dataset (per ref. [68]) by including 2–30 individuals per taxonomic unit sampled from 94 geographic localities representing the majority of US counties, Canadian provinces, and Mexican states for which tissue samples exist (Fig. 2a). We then employed restriction-site-associated DNA (RAD)-sequencing, which offers a reduced representation of the genome that can be mined for thousands of single nucleotide polymorphisms (SNPs) among individuals[69,70].

We extracted whole genomic DNA from each sample using a Qiagen DNeasy Blood and Tissue Extraction Kit and prepared RAD-libraries according to ref. [71] (protocol v.2.3 downloaded from dryad: https://datadryad.org/stash/dataset/doi:10.5061/dryad.m2271pf1). Briefly, we digested whole genomic DNA with two restriction enzymes (*EcoRI* and *Mse1*), ligated adapter sequences with unique barcodes (8–10 bp) for each individual (provided in Source Data file), performed PCR amplification (see Table S2 for primers), and then performed automated size selection of 300–400 bp fragments (Sage Science Blue Pippen). We split paired geographic samples between the two libraries such that all taxa were represented in each library of 96, pooled individuals. Samples were additionally randomized on each plate during library preparation. Libraries were sequenced on separate flow-cell lanes of an Illumina HiSeq 4000 at UC Berkeley's V.C. Genomics Sequencing Lab, yielding over 300 million, 100-nt single-end reads per lane.

We demultiplexed reads, retained reads with intact *EcoRI* cut sites, removed inline barcodes and adapters, and performed quality filtering (removed Phred score <10) using *process_radtags* in STACKS v.2.1[72]. This resulted in final reads of $\mu = 92$ bp in length and 2.05 million reads per individual. We removed three individuals in each lane that failed to sequence (<100,000 reads per individual, comprising five *J. hyemalis* and one *J. insularis*). We used bwa mem[73] to align reads to the *J. hyemalis* genome[37], which we downloaded from NCBI (Accession GCA_003829775.1). An average of 91% of reads mapped and mapping success did not differ among *Junco* species.

We executed the STACKS pipeline to call SNPs and exported one SNP per locus for loci present in all five *Junco* species in vcf format (*populations: -p 5 --write-random-snp*), resulting in 162,856 SNPs. We further filtered the dataset using vcftools v.0.1.16[74]. We removed sites with mean depth of coverage across all individuals <5 (*--min-meanDP*) and >50 (*--max-meanDP*), minimum minor allele count <3 (*--mac*), more than 50% missing data (*--max-missing*), and characterized by indels (*--remove-indels*). We then removed five *J. hyemalis* individuals with >60% missing data (*--remove*). Finally, we removed sites with >5% missing data (*--max-missing*) and filtered three sites that departed from Hardy–Weinberg equilibrium assessed by species ($p < 0.001$), resulting in 22,006 biallelic SNPs across 181 individuals. We exported the dataset in plink.raw format for downstream analysis.

*Genotype-environment association analyses.* To determine if environmental variation corresponded to allelic variation across *Junco*, we employed the SNP dataset in an RDA. RDA is a multivariate ordination technique that has been used to identify multiple candidate loci and several environmental predictors simultaneously[35,36]. Because RDA requires no missing data, we first imputed data for $n = 68,010$ missing sites (1.7% of total sites) using the most common genotype for the species at each site. We then quantified population genetic structure using a PCA of the imputed SNP data with the R package ade4 v.1.7-15[75]. We downloaded 19 spatially interpolated climatic variables aggregated from 1970–2000 corresponding to the geographic origin of each specimen from the WorldClim database[66] at a resolution of 2.5′ using the R package raster v.3.3-13[67]. We centered and standardized climate variables by centering and dividing by two standard deviations[65]. We excluded highly correlated variables ($r ≥ 0.70$) resulting in the retention of eight variables: mean diurnal temperature range (BIO2, a measure of monthly temperature variation), temperature annual range (BIO7), mean temperature of the wettest quarter (BIO8), mean temperature of the driest quarter (BIO9), mean temperature of the warmest quarter (BIO10), precipitation of the driest month (BIO14), precipitation of the warmest quarter (BIO18), and precipitation of the coldest quarter (BIO19). We used these eight climatic variables as constraining variables in RDAs executed with the package vegan v.2.5-6[76]. We performed a partial RDA conditioned on background population structure summarized as PC1 and PC2 from the PCA. We

tested for, but did not find, multicollinearity among predictor variables (variance inflation factor <5 for all variables). We assessed the significance of the RDA model and of the constrained axes using an ANOVA-like permutation technique[77] in vegan at $p ≤ 0.05$ ($n = 999$ and $n = 299$ permutations, respectively). We used variance partitioning[78] to quantify the proportion of allelic variation explained by each climatic variable with the function *varpart*.

### Acclimation experiments

*Population sampling for acclimation experiments.* We combined information gained from a literature search, eBird sightings, and expert opinion to identify five focal populations for phenotypic sampling that (1) were likely to be nonmigratory, (2) represented different morphological subspecies, and (3) maximized variation in annual temperature range within the United States. These populations include *J. h. aikeni* of the Black Hills, a coastal population of *J. h. shufeldti*, a highland population of *J. h. dorsalis*, a sky island population of *J. p. palliatus*, and a well-studied, urban population of *J. h. thurberi*[79]. However, it is possible that some of these populations exhibit seasonal, altitudinal migrations within their geographic area of residence, e.g., *J. p. palliatus*[80].

We captured ≤25 adult individuals from each focal population. Capture periods differed for each population in order to increase the likelihood that targeted individuals were resident year-round, as well as due to time and permitting constraints. For instance, one partially migratory population (*J. h. aikeni*) with distinct morphological features was caught in the winter to increase the likelihood that the individuals used were nonmigratory. The other four populations, which bred in areas where other, morphologically similar juncos overwinter, were captured in the breeding season when other subspecies were not present. Specifically, *J. h. shufeldti* ($n = 20$) were captured 14–15 July 2018 in Coos and Douglas Counties, OR; *J. p. palliatus* ($n = 24$) were captured 27 July 2018 in Cochise County, AZ; *J. h. dorsalis* ($n = 25$) were captured 30–31 July 2018 in Coconino County, AZ; *J. h. aikeni* ($n = 15$) were captured 6–9 March 2019 in Lawrence County, SD; and *J. h. thurberi* ($n = 20$) were captured 22–26 July 2019 in San Diego County, CA. In spite of these differences, capture season did not influence acclimation ability (see below). This work was completed with approval from the US Fish and Wildlife Service (MB84376B-1 to M.S. and MB45239B-0 to T.J.G.), the State of Arizona Game and Fish Department (E19253811 to M.S.), the State of California Department of Fish and Wildlife (13971 to M.S.), the Oregon Department of Fish and Wildlife (108-18 to M.S.), the State of South Dakota Department of Game, Fish, and Parks (19-13 to M.S.), and the Institutional Animal Care and Use Committee at the University of Montana (030-18ZCDBS-052918 to Z.A.C.).

*Acclimation treatments.* Within days of capture, we ground-transported all birds to facilities at the University of Montana where birds were housed individually under common conditions (23 °C with 12 h dark: 12 h light) for ≥8 weeks ($\mu = 63$ days, range = 57–71 days) to minimize the effects of prior acclimatization. We had previously determined that a period of 6 weeks in common conditions is sufficient to reduce variation in metabolic traits among *J. hyemalis* individuals (Fig. S2). Following this adjustment period, we assayed $M_{sum}$ (see below). We allowed birds ~24 h to recover and then randomly assigned individuals from each population into treatment groups and exposed them to either cold (3 °C) or control (23 °C) temperatures. Treatments lasted 21 days in duration. Constant 12 h dark: 12 h light days were maintained for the duration of the experiment, and food and water were supplied ad libitum. The diet consisted of a 2:1 ratio by weight of white millet and black oil sunflower seed, supplemented with ground dog food, live mealworms, and water containing vitamin drops (Wild Harvest D13123 Multi Drops). These experimental conditions were chosen based on previous work in *J. h. hyemalis* exposed to the same temperatures, which revealed substantial increases in $M_{sum}$ over the same duration[21]. At the end of treatments, we euthanized individuals using cervical dislocation.

Eight individuals died during the capture-transport and adjustment periods (one *J. h. dorsalis*, four *J. h. shufeldti*, one *J. h. thurberi*, and two *J. p. palliatus*). Additionally, one *J. h. thurberi* individual exhibited lethargy upon introduction to the cold treatment, died within the first 24 h of cold acclimation, and was removed from analyses. This resulted in a total sample size of $n = 95$ individuals.

*Metabolic assays for acclimation experiments.* We quantified $M_{sum}$ in a temperature-controlled cabinet using open-flow respirometry both before and after acclimation treatments as described above. We measured $M_b$ immediately before each assay. During the post-acclimation $M_{sum}$ trial, we measured body temperature with a pit tag inserted into the cloaca (per ref. [30]) to verify that individuals were hypothermic (body temperature ≤37 °C). $M_{sum}$ trials were conducted using static cold exposure at −5 °C for pre-acclimation measures and −17 °C for post-acclimation measures. Although this experimental temperature difference could have elicited higher post-acclimation $M_{sum}$, we did not find variation in $M_{sum}$ between the two time points in Control birds, suggesting that the two procedures provided similar levels of cold challenge (paired $t$-test: $n = 47$, $t = -1.24$, df = 46, $p = 0.22$). Because trials occurred at various times throughout the day (08:00–19:00), we tested for, but did not find, a linear effect of the trial start time on $M_{sum}$ either before or after acclimation (before: $R^2 = 0.0$, $p = 0.54$; after: $R^2 = 0.0$, $p = 0.94$).

*Climate data for acclimation populations.* We reconstructed the annual thermal regime experienced by a population using interpolated monthly climate data downloaded from the WorldClim dataset[66] with package raster v.3.3-13. We extracted all 19 variables at a resolution of 2.5′ for the site of capture. However, our analysis mainly focuses on BIO7, the temperature annual range variable ($T_{range}$).

*Pairwise genetic distance of acclimated populations.* To quantify genetic differentiation among acclimated populations, we extracted whole genomic DNA from the muscle tissue of acclimated individuals and prepared RAD-libraries as above with an additional PCR step (per protocol v.2.6)[71]. We pooled all 95 individuals into a single flow-cell lane of an Illumina HiSeq X for sequencing at Novogene Corporation Inc. yielding ~1 billion, 150-nt paired-end reads. We demultiplexed reads, removed inline barcodes and adapters, and performed quality filtering (removed Phred score <10) using *process_radtags* in STACKS. This resulted in final reads of $\mu = 147$ bp in length and 7 million reads per individual. We removed six individuals that sequenced poorly (<200,000 reads/individual, comprising one *J. h. dorsalis*, one *J. p. palliatus*, one *J. h. shufeldti*, and three *J. h. thurberi*). We aligned reads to the *J. h. carolinensis* genome with bwa mem (51% of reads mapped). We then sorted, merged, and removed read duplicates with samtools v.1.3[81]. We used the STACKS pipeline to call SNPs from loci present in all five populations and exported one SNP per locus in vcf format resulting in 282,929 SNPs (*populations: -p 5 --write-random-snp*).

We filtered the dataset and estimated patterns of genetic differentiation using vcftools. We removed sites with mean depth of coverage across all individuals <20 (*--min-meanDP*) and >300 (*--max-meanDP*), minor allele frequency <5% (*--min-maf*), and >10% missing data (*--max-missing*), as well as removed indels (*--remove-indels*), resulting in 1069 biallelic SNPs across 89 individuals. Finally, we estimated pairwise $F_{ST}$ among the focal taxa by calculating the weighted Weir's theta[82].

*Analyses for acclimation data.* We first performed a partial mantel test to ascertain that environmental and genetic distances did not covary among our sampling sites. We calculated pairwise genetic distance as $F_{ST}/(1 – F_{ST})$[83]. We estimated pairwise environmental differences among the five sites as the Euclidean distance for five WorldClim variables (after removing redundant variables at $r \geq 0.70$ from the original 19 WorldClim variables). We simultaneously controlled for geographic distance, estimated as pairwise geodesic distance among sampling sites with the package geosphere v.1.5-10[84]. We then employed these matrices of pairwise genetic and environmental distances in a partial Mantel test (conditioned on geographic distance) with the package vegan v.2.5-6[76].

We quantified the effects of our acclimation treatments while simultaneously incorporating population demography using Markov Chain Monte Carlo generalized linear mixed models[45,46] with the package MCMCglmm v.2.32[85]. This allowed us to include pairwise $F_{ST}$ as a random effect in all analyses. Because individual measures were used in these models, the pairwise $F_{ST}$ among two individuals corresponded to that of their respective subspecies. We standardized all continuous predictor variables by centering and dividing by two standard deviations[65]. In all cases, we used default priors and ran the model for 1,000,000 iterations with a burn-in of 10,000 and a thinning interval of 100. We examined the resulting trace plots to verify proper convergence.

We first verified that phenotypic differences did not exist among treatment groups before acclimations began. To do this, we constructed a model to explain variation in pre-acclimation $M_{sum}$ with temperature treatment as the main effect and pairwise $F_{ST}$ as a random effect. We then repeated this procedure to test for differences among treatments in pre-acclimation $M_b$. We constructed similar models to test the effect of $T_{range}$ on pre-acclimation measures of each $M_{sum}$ and $M_b$. We also tested for (but did not find) an effect of capture season (breeding or nonbreeding) on pre-acclimation $M_{sum}$ while including $M_b$ as a covariate (season: posterior $\mu = 0.42$, 95% CI = [−1.48, 2.30], $p = 0.57$; $M_b$: posterior $\mu = 0.45$, 95% CI = [0.01, 0.88], $p = 0.04$).

To evaluate whether environmental variation corresponded with thermogenic flexibility, we constructed a model explaining variation in thermogenic flexibility ($\Delta M_{sum}$; post- minus pre-acclimation $M_{sum}$) with pre-acclimation $M_b$, temperature treatment, $T_{range}$, and a treatment × $T_{range}$ interaction term as main effects and pairwise $F_{ST}$ as a random effect. We tested for (but did not find) an effect of capture season on $\Delta M_{sum}$ while including $M_b$ as a covariate (season: posterior $\mu = 0.09$, 95% CI = [−0.42, 0.58], $p = 0.72$; $M_b$: posterior $\mu = 0.67$, 95% CI = [0.33, 1.03], $p = 6.0 \times 10^{-4}$).

Finally, we calculated CV, (expressed as percentages)[86] for $\Delta M_{sum}$ across cold-acclimated individuals within each population using the package raster. To test whether populations from more variable environments expressed less variability in their flexible response, we then regressed the five CV values on $T_{range}$ for each population using a linear regression. We also repeated this analysis for control-acclimated birds.

## Data availability

Source Data are provided with this paper. Additionally, raw sequence reads have been deposited in the NCBI Sequence Read Archive under accession number PRJNA678344. Environmental data were accessed in R from the Daymet [https://daymet.ornl.gov/] and WorldClim [https://www.worldclim.org/data/bioclim.html] datasets. Source data are provided with this paper.

## Code availability

The scripts used for processing and analysis are deposited on github (https://doi.org/10.5281/zenodo.4968784).

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

## Acknowledgements
We are indebted to the natural history museums that contributed to this project, including the Burke Museum, the American Museum of Natural History, the Bell Museum of Natural History, the Cleveland Museum of Natural History, the Cornell University Museum of Vertebrates, the Field Museum of Natural History, the Louisiana State University Museum of Natural Science, the Museum of Southwest Biology, the Museum of Vertebrate Zoology, the New York State Museum, the Royal Alberta Museum, the San Diego Natural History Museum, the Smithsonian National Museum of Natural History, the University of Alaska Museum, the University of Montana Zoological Museum, and the University of Wyoming Museum of Vertebrates. We are also thankful to the help of Brett Addis, Phred Benham, Jeff Brawn, Kevin Burns, Charles Dardia, Eleanor Diamant, Eric Gulson, Andy Johnson, John Klicka, Thom Nelson, Henry Pollock, Trey Sasser, Rena Schweizer, Nick Sly, Link Smith, Gregory Toreev, Phil Unitt, David Winkler, Chris Witt, Blair Wolf, Cole Wolf, Sally Woodin, Pamela Yeh, Point Loma Nazarene University, the UNM Sevilleta Field Station, the UW-NPS Research Station, the DU Mt. Evans Field Station, and the AMNH Southwestern Research Station, as well as the Cheviron lab for feedback on an earlier version of this manuscript. This work was supported by generous funds from the American Museum of Natural History Chapman Fund, the American Philosophical Society Lewis and Clark Fund, the Explorers Club, the Illinois Ornithological Society, the Nuttall-Ornithological Society, Sigma Xi, the Society for Integrative and Comparative Biology, the Society of Systematic Biologists, the University of Illinois Graduate School, the University of Illinois School of Integrative Biology, and the Wilson Ornithological Society (to M.S.); and the University of Montana (startup to Z.A.C). M.S. was supported by the National Science Foundation Graduate Research Fellowship Program, P.E.O. International, and the University of Montana Graduate School Bertha Morton Fellowship.

## Author contributions
M.S. and Z.A.C. conceived of the study; M.D.C. and N.R.S. helped perform field work; D.L.S. performed field measurements in AZ and SD; D.K.E. helped perform field measurements in CO and WY; T.J.G. provided sampling permits; M.S. performed all other data collection and analyses and drafted the manuscript; all authors contributed edits to the manuscript.

## Competing interests
The authors declare no competing interests.
