## [Peer Review File · Nature Communications]

REVIEWER COMMENTS

Reviewer #1 (Remarks to the Author):

The authors present a thorough and interesting study exploring how environmental temperature heterogeneity is related to cold tolerance and genetic variation in *Juncos*.

I feel the study is methodologically sound and am on board with the general aims and genetic conclusions. However, I have some concerns about the reach of the phenotypic conclusions. I am currently not understanding how the Msum values are justified as measures of phenotypic flexibility. I think further explanation of this link, or alternately an analysis of variation in Msum, is needed to support the current MS conclusions.

Currently, I think the emphasis in the abstract, results, and discussion should rather be on how cold tolerance is related to temperature heterogeneity. Specifically, in the abstract I would rephrase point 2 to read: exhibit intra-specific variation in their cold tolerance in the laboratory that correlates with heterogeneity in their native thermal environment. Also, as the manuscript is currently written I cannot support the concluding sentence about flexibility, as I am not convinced flexibility is actually analysed in the current version.

In my understanding, Msum is just a measure of the maximum aerobic capacity of an individual when pushed to the limit of cold tolerance. It doesn't really tell you about physiological or phenotypic 'flexibility' of an individual. From my understanding of the study methods, in the first section on geographic variation in thermogenic performance- the authors assayed Msum once per individual and find correlations between Msum and the temperature range they are naturally exposed to. I am on board with these findings. However, the authors then go on to conclude that this demonstrates that "populations may differ in their physiological flexibility and that variation in the temperature range across their distribution may play an important role in shaping this flexibility." I do not understand how you use a single measure of the limit of cold tolerance as a proxy for flexibility? From the statistical methods section and Figure 2 results, I believe Msum is the response variable for these models. But from the way you phrase your questions and conclusions, I feel like it would be more appropriate to use variation in Msum to make this statement. I think what would really answer this question is an analysis that leads to a figure such as the attached 'alternative figure 2' jpeg file.

I used the data presented in the current figure 2 to pull out rough estimates for each population and plotted them. This type of analysis is what would demonstrate that there is more flexibility (ie variation in Msum amongst individuals) in populations with greater temperature variation. Alternatively, there needs to be a better explanation of why and how the Msum variable you use demonstrates physiological flexibility.

For the third part of the study, on "Flexible responses to temperature acclimation treatments", I have a similar problem. How did you decide that a larger deltaMsum was indicative of more physiological

flexibility? Could it not be said that a bird that maintains the same Msum at different temperatures is actually the more flexible individual, since the energy requirements are different, thus their energy expenditure capacity is more variable? Also, I am again missing an analysis of variation in Msum within populations. Metabolic scope (Msum minus basal metabolic rate) would be a better measure here if you have the data as it takes into account the total metabolic envelope from basal metabolic rate at the low end to Msum at the top end. This would provide a better measure of an idea of an individual's 'flexibility'. If using Msum, it also depends on the temperature at which it's reached. Did you look at or analyse the temperature at which a plateau in metabolic rate was found when doing the sliding cold exposure?

Reviewer #2 (Remarks to the Author):

This paper contains fantastic, important data demonstrating an association between metabolic performance and environmental variability. The experimental design is brilliant, for example, by focusing on resident populations for the acclimatization studies, and extensive genetic analysis to control for relatedness. It deserves to be widely quoted and would be worth publishing in Nature Communications.... Eventually.

At present much needs to be done before I can give a proper assessment; the paper consists of many statements that are not connected, and it is very difficult to know how the analyses were done:

-This is a paper about thermal tolerance, and the introduction seems quite off topic. Sticklebacks and bindweed, and even any discussion at all of one-shot plasticity seem quite irrelevant

-The connection between the remarkable work on thermogenic performance and long-term acclimation experiments is never weaved together, but this would be the way to organize the introduction and discussion. How would they be connected under different hypotheses?

-The genetic predictions, and the terminology used is very unclear. I think the only place to use the genetics is in the RDA analysis. (why are you looking at outlier SNPs, how does a sudden reference to Friis et al fit in?)

-Figures and analysis need improvement. The genetic ones are dominated by a few outliers. More descriptive axis labels would help, as would units. Can't you place labels on the figures like this: Maximum instant O2 consumption (Msum)

-The analyses and technical details need to be explained throughout. For example, Fst is a 'random effect' in MCMCglmm but it seemed entries were of individuals. Elsewhere Fst/(1-Fst) is used cf. Fst, without explanation.

-I had to go miles to find that Msum is quantified as: "the highest instant O2 consumption averaged over a 5-min period" and it should be defined early. Is Mb defined anywhere???

-Results and discussion should be separated, with conclusions part of the discussion, and methods moved out of the results.

A few general comments on the introduction below, rest are on the manuscript, which I am returning.

In the evolutionary literature the term labile traits (Scheiner 1993) is much more commonly used than phenotypic flexibility.

See Figure 2 of Lande 2014 for the demonstration of exactly what you are saying:

“The expected plasticity that evolves in labile characters depends on a ratio: the cost of plasticity relative to the strength of stabilizing selection on the character, divided by the product of environmental variance and environmental predictability averaged over the developmental time lag”

This is a much more relevant paper than (19).

my reading of the Overgaard et al differs from that of the authors:

I wrote in an unpublished review: Experiments by Overgaard et al. (2011) illustrate how various forms of plasticity contribute to variation in the tolerance curve across climatic regimes. These authors compared five widespread Australian *Drosophila* species with five restricted to the tropics, knocking down each with an abrupt lowering of temperature. Indicative of underlying labile traits, presumably physiological, widespread species were both knocked down and recovered from knockdown at lower temperatures than tropical species. Indicative of one-shot plasticity, if eggs were allowed to develop in a cold regime, individuals were knocked down and recovered from knockdown at lower temperatures than when eggs were reared at preferred temperatures; this was especially true for the widespread species. Responses to hot regimes were less marked, yielding a net effect of greater tolerance for the widespread species.

Reviewer #3 (Remarks to the Author):

This paper tests the hypothesis that plasticity in thermogenic abilities varies with environmental heterogeneity – more variable environments promoting increased plasticity. The authors address these questions with a three-pronged approach that is quite impressive in scope. First, by measuring metabolic responses to an administered cold stress in the field; second, by measuring genetic variation (RADseq) across populations/species in the genus *Junco*; third, by experimentally assessing thermal acclimation across population in a common garden experiment in captivity. This study is important and

timely. Furthering our understanding of how species and populations diverge in thermal acclimation is the foundation needed for predicting species persistence into the future.

These are the main results:

1. Populations of Juncos vary in thermogenic flexibility, suggesting that local environments shape phenotypic plasticity.
2. Annual temperature range explains the most genetic variation of the environmental metrics measured.
3. In a common garden study, individuals from more variable environments were able to mount a more robust thermogenic responses as measured by an increase in M-sum (peak metabolic rate under cold exposure).

Despite these strengths, I have three major concerns, chiefly:

- (a) Birds are sampled from breeding and wintering seasons. Seasonal differences are well documented in Juncos. This does not appear to be balanced within species/morphotypes or controlled for statistically.
- (b) It is unclear how the museum samples were handled when addressing how genetic variation relates to climate - What time period of weather were the museum data correlated with, since some samples date back to the 1980s? And how does that contrast with the climate analyses? I'm unclear on the degree of matching here – do the genetic samples and environmental data cover the same time period, or are they sampled generations apart?
- (c) The writing and structure of the paper was, at times, hard to follow for a reader not yet familiar with the project. This is partly driven by the journal's constraints on the order of presentation and compounded by jargon that is not explained and/or explained too late (Msum, acclimatization, in situ, that the focus on the project is cold tolerance, etc.).

I have elaborated on these points and other more minor issues below (by line number).

15: Somewhere in the abstract, please state that your study is focused on thermal challenges in a cold capacity. This was not evident until the final paragraph of the introduction.

first two paragraphs of the introduction: The framing of the paper put a lot of emphasis on developmental plasticity, and this was not returned to in the discussion, suggesting its not a relevant framing. I recommend the authors better use this space to set up the proposed work. I agree with the authors' argument that much of past research on plasticity has focused on development, but there is quite a lot of work on adult or moment-to-moment plasticity, which could be incorporated here too. I especially encourage more citation of non-bird or even non-animal work.

52: "Unlike developmental plasticity, phenotypic flexibility..." Are the authors trying to differentiate different temporal scales of plasticity? Is phenotypic flexibility meant to signify something different from phenotypic plasticity?

61 & 69: These statements of novelty seem overstated, given past work in plants or ectothermic animals. I'm thinking for example of Lowry et al.'s (2019) PNAS paper on switchgrass across most of

North America or Whitehead's work on killifish.

62: Please clarify how this study accounts for "non-independence among populations (due to shared common ancestry and ongoing gene flow)". I assume the authors are simply advocating for appropriate phylogenetic methods, but please clarify, as some readers might also think that, say, spatially explicit statistical models would account for this as well.

82: To what degree does the lack of genetic variation in Junco subspecies undermine the premise of this work? For a generalist audience, it would help to understand the scope of genetic variation seen here, and how it compares to other work, though this becomes more relevant later in the manuscript for those not already familiar with the previous studies on population genetics in Junco from the Mila research group.

90: For the generalist reader, I recommend defining Msum more clearly by stating what it stands for: summit metabolism.

91-3: Is "thermogenic capacity" synonymous with thermogenic performance? Please clarify. This comes up elsewhere in the paper too, related to thermogenic flexibility.

104: "in situ" is a broadly used term in biology, and it means completely different things depending on the discipline. I understand that "in the field" is not exactly accurate (or is it? I'm unclear on how birds were housed when pulled from the field in the first study). However, this made it hard to understand the results and discussion without also reading the methods, which come much later. Please clarify the language, so the reader can understand key points as they go.

107: "Laboratory acclimation" is also unclear. The degree of jargon or otherwise undefined terms in this final paragraph of the introduction made it hard for me to know exactly what was done here, until I also went and read the methods.

section beginning 118: I appreciate that everything cannot be included in the main text, especially with word limits and an integrative 3-part study like this... but knowing generally that this occurred in certain seasons is important. Please include that in this paragraph.

131: Result is hard to interpret without some sort of even balanced sampling across seasons, even within a morphotype. I also had to spend a lot of time reading through SI data tables to match species/subspecies with seasonality and location. I am concerned that there appears to be imbalance in the seasonal sampling across populations. How can we compare species or morphotypes if some are sampled only in winter and other only in summer?

131: Switching between common and scientific names is disorienting. Please choose one to use consistently throughout the figures and tables and corresponding text.

137-138: This is a compelling result, but I am still worried about the differences of seasonality in the birds were sampled.

Figure S1: I would really appreciate your Figure S1 map of the sub-species ranges included in the main text if there is room. Or perhaps there is some way to convey this alongside the existing map? A broad audience may not be able to retain the geographical areas of each species/subspecies and therefore may not be able to interpret your results without frequently flipping pages or referring to your supplemental. Consistent terminology may help to resolve some of this.

157: Clarify goal of the RDA. Why exactly is this needed beyond the PCA? And, how meaningful is it for each dimension to explain a maximum of 0.6% of the variation (R^2 , conditioned = 0.0060 for temperature range).

161: “conditioned” and “unconditioned” are not completely clear yet. Because of the order of this paper/this journal, you need to walk the reader through your terminology before your methods.

187-188: Please clarify this sampling regime. I believe the authors purposefully selected non-migratory animals for this third study, to make sure they could know their temperature history. This seems like a biased sample within the Junco. Genetic variation (that stems from temperature variability) must be different in populations that experience extreme climatic variability. If the authors can show that the non-migrants have unbiased variability, then this study design would seem more appropriate. Ultimately, this tempers my enthusiasm for linking the genetic and trait-based portions of this paper, because the climate regimes shaping a migrant sampled in Canada (who winters in the US somewhere) may be totally different from the climatic regimes shaping a bird who spends in whole life just in the Black Hills.

216-7: I’m not convinced we can conclude environmental heterogeneity is an important selective force. I need to know more to be able to assess this claim. How much variation are we explaining, and is this an important ‘amount’? What is the scope of these R^2 ? Is the # of SNPs high or low? Is this a real but small effect? Please contextualize with other work.

219-221: It may be true that these correlative studies have not been done in ectotherms, but there are studies that have looked at plasticity differences in ectotherms and plants resulting from environmental differences. The novelty of this study seems overstated in light of this.

220, 223, 226, 230, 232, etc: Please choose either “plasticity” or “flexibility.”

257: Explain why birds from wintering and breeding grounds can be comparable. I have concerns with comparing birds of different physiological states without controlling for seasonality.

267: Did you test for an effect of sampling time (within 48h vs within 24h)? This seems like a wide time frame knowing how much stress hormones should interact with metabolism, particularly for a species that has been widely studied for its variation in corticosteroid reactivity (which can vary among populations and seasonally). Please clarify so that the reader has more confidence in there being stable differences among populations in their peak metabolic capacity.

274: "Static cold exposure:" I assume this means that you exposed birds to cold suddenly. Did the authors present how the birds were housed (what temp) during the 24-48 after capture?

288: Define "acclimatization" or use a simpler term like "recent thermal experience" or "recent thermal history".

300: Please explain more about the climatic windows. 7 to 14 days is quite a wide range. Were some windows days 0 to 7 prior to capture and others were 0-14 days prior to capture? Or did you measure days 7-14 prior to capture? In this case, why were the most recent 7 days prior to capture thrown out? I was unclear on exactly how this was done from the main text.

349: Please explain why sites that departed from Hardy-Weinberg equilibrium were filtered out. Also, minor typo: I believe it's 'berg', not 'burg'.

354: This topic sentence would be helpful on line 155. Consider moving.

359: What was the time frame that the weather data was averaged across (how many months) for the genotype-env association analyses? I.e., 12 months before each muscle sample was collected? How was this managed since the museum tissues were sampled across decades?

Related to this point, is historical time balanced by lat-long in the sampling regime? Is season balanced within in the sampling regime? I understand that RADseq is independent of when an animal is captured, but this does matter when we decide what climate to assign to that bird. This is particularly important considering that some museum samples are from a very long time ago (decades), before some of these populations even bred in their current locality.

362-366: Consider renaming environmental variable to more intuitive abbreviations.

366: How do these environmental metrics relate to migratory status? Please clarify: are you using climate data from the place where the bird was captured?

368: Does "partial RDA conditioned on background population structure" mean it accounts for genetic differences and phylogeny? Using something external or this specific dataset?

403: "Erase" is an overstatement. Please soften this term.

404-6: Please elaborate how this result shows that birds are not making seasonal adjustments. If Delta AIC is under two, then we cannot reject this as a parsimonious model, though I recognize that this p value suggests this variable is not meaningful here.

415-416: Do the species differ in whether they were in breeding condition, suddenly experiencing a lengthening, or suddenly experiencing a shortening in day length based on the switch from capture site to 12:12 in the lab? I cannot help but think about how much is changing within the birds' physiology during that adjustment. How was season treated in the aforementioned model in my last comment?

422: Does the regression of brood patches and cloacal protuberances indicate that after the adjustment period there were no biological differences between breeding and wintering individuals? The timing here is unclear and incredibly important. Are these birds regressing, recently regressed, or just prior to recrudescence?

436: I am not convinced we can reject the idea that birds improved in their cold tolerance (or vice versa) because the challenge changed.

Figure 1: color dots are hard to parse out, consider enlarging the figure or adjusting the format for more clarity so that multi-colored dots are not obscured.

Figure 2: I prefer figure captions to not require reading the text to understand. Please define Msum and Mb. Please include the seasonality (breeding/winter) and sampling location in this figure for each of the subspecies/species. If I got it right from looking through the SI material, yellow-eyed birds are only sampled in spring, and gray-headed birds are only sampled in non-breeding. Please also restrict the trend line to extend only through the range of data.

Figure 3: I had to reference supplemental to understand this figure. I'm not convinced that the PCA plots (a and b) add much, except to reiterate that there really isn't much genetic differentiation among most samples.

Figure 4: This is a very cool result! Please include location and seasonality of each population. *J. h. aikeni* were the only birds sampled in March (all others in July). Could these birds be already primed to respond to cold? But the general result still holds even with omitting the *J. h. aikeni* data, so I suspect this cannot explain the pattern.

Supplemental: Why does Table S3 present 5 different models with different reference-morphotypes? Please clarify.

REVIEWER COMMENTS

Reviewer #1 (Remarks to the Author):

The authors present a thorough and interesting study exploring how environmental temperature heterogeneity is related to cold tolerance and genetic variation in Juncos.

I feel the study is methodologically sound and am on board with the general aims and genetic conclusions. However, I have some concerns about the reach of the phenotypic conclusions. I am currently not understanding how the Msum values are justified as measures of phenotypic flexibility. I think further explanation of this link, or alternately an analysis of variation in Msum, is needed to support the current MS conclusions.

We thank you for taking the time to review our manuscript and provide feedback. We appreciate your support of the article and have worked to incorporate your comments on the analysis of our field measurements.

Currently, I think the emphasis in the abstract, results, and discussion should rather be on how cold tolerance is related to temperature heterogeneity. Specifically, in the abstract I would rephrase point 2 to read: exhibit intra-specific variation in their cold tolerance in the laboratory that correlates with heterogeneity in their native thermal environment.

We have refocused the manuscript to place more emphasis on thermal tolerance overall; however, we have not focused on cold tolerance specifically. While thermogenic capacity and cold tolerance are correlated, we did not quantify cold tolerance. Cold tolerance is generally quantified by exposing individuals to different temperatures and assessing the proportion of individuals that become hypothermic at each temperature or exposing individual birds to a standardized cold exposure to measure time to hypothermia.

Also, as the manuscript is currently written I cannot support the concluding sentence about flexibility, as I am not convinced flexibility is actually analysed in the current version. In my understanding, Msum is just a measure of the maximum aerobic capacity of an individual when pushed to the limit of cold tolerance. It doesn't really tell you about physiological or phenotypic 'flexibility' of an individual. From my understanding of the study methods, in the first section on geographic variation in thermogenic performance- the authors assayed Msum once per individual and find correlations between Msum and the temperature range they are naturally exposed to. I am on board with these findings. However, the authors then go on to conclude that this demonstrates that "populations may differ in their physiological flexibility and that variation in the temperature range across their distribution may play an important role in shaping this flexibility." I do not understand how you use a single measure of the limit of cold tolerance as a proxy for flexibility? From the statistical methods section and Figure 2 results, I believe Msum is the response variable for these models. But from the way you phrase your questions and conclusions, I feel like it would be more appropriate to use variation in Msum to make this statement. I think what would really answer this question is an analysis that leads to a figure such as the attached 'alternative figure 2' jpeg file. I used the data presented in the current figure 2 to

pull out rough estimates for each population and plotted them. This type of analysis is what would demonstrate that there is more flexibility (ie variation in Msum amongst individuals) in populations with greater temperature variation. Alternatively, there needs to be a better explanation of why and how the Msum variable you use demonstrates physiological flexibility.

This is a key point, and we regret that this was not clear in the original manuscript. The reviewer is correct in understanding that we measured each individual from the field study only once and, thus, these are not direct measures of flexibility. We do, however, think the pattern we observe is suggestive of differences in flexibility among populations because of differences in the slopes (i.e., reaction norms) among populations. Moreover, the relationship that the reviewer would like us to add (variation in Msum within a population ~ Temperature Range) actually already exists in our field results via the Taxon:Temperature Range interaction term. We elaborate on these points below.

Individuals within a population exhibit variation in Msum that corresponds with variation in daily temperature range, such that individuals measured at the same location days apart often exhibited differences in Msum. It is unlikely that these differences simply reflect static inter-individual differences in Msum alone. If it did, we would not expect a relationship with temperature range within a site. However, subspecies do show positive correlations between Msum and temperature range, suggesting that individuals within a population are responding to temperature variation. In this way, we view the relationships between Msum and temperature range, plotted in Figure 1 (formerly Fig. 2), as reaction norms of flexibility. Not all subspecies exhibit the same slope for this relationship, which leads us to conclude that subspecies likely differ in their flexibility. We have expanded upon our reasoning in the Discussion to make this clear (Lines 196-204).

We agree that the focus for the analysis of the field data is on the variation in Msum. The analysis that we conducted draws power from the large number of individuals and environmental variation that we captured in this dataset. We therefore do not want to condense the variation to recreate the analysis/plot that the reviewer suggests. However, this result exists already, in that the interaction term Taxon:Temperature Range demonstrates that the relationship between Msum and temperature range differs by taxa.

We did, though, perform an additional analysis to address the reviewer's comment and created a plot very much related to what they have suggested (Fig. S1). Here we did not use the variation in Msum *per se*, but rather the slopes describing the relationship of Msum and temperature range. We tested for a correlation between the taxon-specific reaction norms (i.e., slope; as indices of thermogenic flexibility) and the native temperature range that each taxon experiences (using the long-term climate dataset from WorldClim). Thus, in this analysis we have one estimate of each flexibility and thermal heterogeneity per taxon. Our result is not wholly unequivocal, in that we do not find a correlation between them; however, if we remove one of the five populations (*J. h. mearnsi*), we find a very strong correlation ($R^2 = 0.90$). *J. h. mearnsi* differs from the other populations in that the slope of the relationship between flexibility and thermal heterogeneity is quite flat. We now present this analysis in the Results (Lines 129-135) and elaborate upon this finding in the Discussion (Lines 204-218).

For the third part of the study, on “Flexible responses to temperature acclimation treatments”, I have a similar problem. How did you decide that a larger ΔM_{sum} was indicative of more physiological flexibility?

We apologize that our reasoning was not more transparent. Because flexibility is measured as a change in a trait value, a greater magnitude of flexibility would also correspond to a greater magnitude of change in the trait value. We therefore define the degree of flexibility in M_{sum} as the change in M_{sum} with acclimation. We have added language to make this clearer in the Introduction (Lines 100-101).

Could it not be said that a bird that maintains the same M_{sum} at different temperatures is actually the more flexible individual, since the energy requirements are different, thus their energy expenditure capacity is more variable?

No, we do not agree that this would be correct, unless we are misunderstanding what the reviewer is trying to illustrate. The energy requirements are different and yes, their energy expenditure (measured as their oxygen consumption) changes. Maintaining the same M_{sum} at different temperatures would require differences in thermal conductance ($M_{sum} = C[T_b - T_a]$), which is related to insulatory capacity rather than physiological flexibility, the latter of which is the focus of the present study. We are specifically interested in flexibility in M_{sum} , thus the change in M_{sum} (i.e., change from pre- to post-acclimation) is of primary importance. If the reviewer is suggesting that after acclimation, individuals could generate more heat while using less oxygen, this would instead point to large differences in efficiency and thus flexibility at lower levels of biological organization. This would be very interesting but has not yet been demonstrated in birds, nor would it show “their energy expenditure capacity is more variable.” As we state, the existing literature shows that birds increase M_{sum} in the winter to cope with temperature stressors, and we have therefore focused on these changes in M_{sum} as a measure of flexibility.

To provide further evidence that the “inflexibility” observed in two of the populations in the acclimation experiment did not result from a failure to sufficiently stress these populations (such that phenotypic change was not necessary to meet the cold exposure challenge), we have included an additional statement that all individuals were hypothermic at the end of the M_{sum} trial (Lines 250-252).

Also, I am again missing an analysis of variation in M_{sum} within populations. Metabolic scope (M_{sum} minus basal metabolic rate) would be a better measure here if you have the data as it takes into account the total metabolic envelope from basal metabolic rate at the low end to M_{sum} at the top end. This would provide a better measure of an idea of an individual's 'flexibility'.

Metabolic scope would provide a measure of an individual's instantaneous flexibility, but here we are specifically interested in their acclimation capacity. We have previously shown that junco resting metabolic rate does not acclimate with cold exposure, despite increases in M_{sum} in the cold (Stager et al. 2020, *J. Exp. Biol.*). Moreover, including BMR data here would not provide

parity with the field data (for which we do not have BMR data) and we therefore think it would introduce confusion.

If using Msum, it also depends on the temperature at which it's reached. Did you look at or analyse the temperature at which a plateau in metabolic rate was found when doing the sliding cold exposure?

We did not use sliding cold exposure in the third part of the study; we used static cold exposure (Lines 496-497). Previous studies, though, have shown that the Msum values measured with both methodologies are similar (Swanson et al. 1996, *J. Thermal Biol*).

Reviewer #2 (Remarks to the Author):

This paper contains fantastic, important data demonstrating an association between metabolic performance and environmental variability. The experimental design is brilliant, for example, by focusing on resident populations for the acclimatization studies, and extensive genetic analysis to control for relatedness. It deserves to be widely quoted and would be worth publishing in Nature Communications.... Eventually.

We thank you for taking the time to review our manuscript and provide such detailed feedback. We appreciate your support of the article and have worked to incorporate your comments on framing, clarity, and integration across the three datasets.

At present much needs to be done before I can give a proper assessment; the paper consists of many statements that are not connected, and it is very difficult to know how the analyses were done:

-This is a paper about thermal tolerance, and the introduction seems quite off topic. Sticklebacks and bindweed, and even any discussion at all of one-shot plasticity seem quite irrelevant

We have completely rewritten the Introduction and have removed all references to developmental plasticity.

-The connection between the remarkable work on thermogenic performance and long-term acclimation experiments is never weaved together, but this would be the way to organize the introduction and discussion. How would they be connected under different hypotheses?

We now list explicit predictions in the Introduction and spend more time weaving together the various components of the study in both the Introduction and the Discussion.

-The genetic predictions, and the terminology used is very unclear. I think the only place to use the genetics is in the RDA analysis. (why are you looking at outlier SNPs, how does a sudden reference to Friis et al fit in?)

We have used genetic data in both the RDA and in the analyses regarding the acclimation study. We have removed the outlier analysis and reference to Friis here.

-Figures and analysis need improvement. The genetic ones are dominated by a few outliers. More descriptive axis labels would help, as would units. Can't you place labels on the figures like this: Maximum instant O2 consumption (Msum)

We have greatly modified the figures, the figure captions, and the axis labels in accordance with these comments and those from the other reviewers.

-The analyses and technical details need to be explained throughout. For example, Fst is a 'random effect' in MCMCglmm but it seemed entries were of individuals. Elsewhere $F_{ST}/(1-F_{ST})$ is used cf. Fst, without explanation.

We have added additional details throughout the text.

In this particular case, individual values were used in the MCMCglmm, so each entry in the matrix corresponded to the relationship between two individuals and reflected the pairwise F_{ST} for their respective populations. We have added this explicit detail in the Methods (Lines 542-543). In the partial mantel test, though, we calculated genetic distance as $F_{ST}^* = F_{ST}/(1-F_{ST})$ according to Rousset 1997 (see comment below).

-I had to go miles to find that Msum is quantified as: "the highest instant O2 consumption averaged over a 5-min period" and it should be defined early. Is Mb defined anywhere???

We have added the quantification of Msum to the Introduction (Line 81-83). Body mass (Mb) continues to be defined upon its first use (Line 119).

-Results and discussion should be separated, with conclusions part of the discussion, and methods moved out of the results.

We have separated the Results and Discussion as suggested. We tried to include brief descriptions of the Methods in the Introduction because all three reviewers pointed out that it was difficult to know what we had done. However, because it is the journal's policy to put the Methods at the end of the manuscript, in a few instances we felt some explanation was still necessary in the Results (e.g., Lines 144-149). These short passages are necessary to give the reader context for interpreting the results that follow.

A few general comments on the introduction below, rest are on the manuscript, which I am returning.

In the evolutionary literature the term labile traits (Scheiner 1993) is much more commonly used than phenotypic flexibility. See Figure 2 of Lande 2014 for the demonstration of exactly what you are saying: "The expected plasticity that evolves in labile characters depends on a ratio: the cost of plasticity relative to the strength of stabilizing selection on the character, divided by the product of environmental variance and environmental predictability averaged over the developmental time lag." This is a much more relevant paper than (19).

We politely disagree with the reviewer that “labile traits” is the more commonly used term. A quick Web of Science search (performed on March 9, 2021) finds that “labile traits” returns 578 results, whereas “phenotypic flexibility” returns 1385 results. We are quite familiar with this literature and the tangled nest of terms associated with plasticity and flexibility, and therefore do not wish to contribute to further confusion. Thus, we have chosen to use the more salient term (phenotypic flexibility), but have added the Lande 2014 citation, as well (Line 35).

my reading of the Overgaard et al differs from that of the authors:

I wrote in an unpublished review: Experiments by Overgaard et al. (2011) illustrate how various forms of plasticity contribute to variation in the tolerance curve across climatic regimes. These authors compared five widespread Australian *Drosophila* species with five restricted to the tropics, knocking down each with an abrupt lowering of temperature. Indicative of underlying labile traits, presumably physiological, widespread species were both knocked down and recovered from knockdown at lower temperatures than tropical species. Indicative of one-shot plasticity, if eggs were allowed to develop in a cold regime, individuals were knocked down and recovered from knockdown at lower temperatures than when eggs were reared at preferred temperatures; this was especially true for the widespread species. Responses to hot regimes were less marked, yielding a net effect of greater tolerance for the widespread species.

As the reviewer suggested, we have removed the discussion of developmental plasticity from the Introduction and therefore no longer cite Overgaard et al. 2011.

In order to address the comments that Reviewer 2 included on the attached document we have copied them below:

Line 18: (Highlighted “integrated”) Perhaps you need to emphasize (1) and (2) and use genetics as a control for these.

This comment was not clear to us. The genetic data serve in the acclimation analysis, where we have controlled for F_{st} among populations with our MCMCglmm models. They also serve in an independent analysis for detecting genotype-environment associations.

Line 25: (Highlighted “harbor genetic variation that also correlates with temperature heterogeneity”) Where did you show this?

This result was shown in our genotype environment association analysis. We have rewritten portions of this passage to better highlight this result (Lines 144-154)

Line 30: (Highlighted “multiple”) Alternative? I would probably drop all reference to one-shot plasticity, e.g. sticklebacks etc. it seems that thermal tolerance is what you should be focused on, and there are relevant refs such as JM Sunday in Nature Climate Change

We have dropped all references and discussion of developmental plasticity as the reviewer suggested.

Line 33: (Highlighted “standing genetic variation in plastic”) Seems irrelevant to this paper.

We have removed this statement.

Line 35: (Highlighted adaptive plasticity should increase fitness) Tautology: if adaptive, increases fitness???

We have removed the word “adaptive” from this statement.

Line 38: (Highlighted “but most”) Not if you include behaviors

We have removed this statement.

Line 51: (Highlighted “phenotypic flexibility”) Labile traits, including those that determine thermal tolerance...

We have removed this statement.

Line 55: (Highlighted “behavioral traits”) Isn’t this by definition?

We have removed this statement.

Line 60: (Highlighted “and none have...variation.”) This may be true, but it is a technical issue. It is cool you have done this and you should emphasize in the discussion, but I would focus on the positive aspects. None have done anything like the thorough study reported here.

We have removed this statement.

Line 65: (Highlighted “we would therefore...as well.”) Why? This is very unclear. Are we talking about within population? How would this aid lability?

We have removed this paragraph.

Line 66: (Highlighted “environmentally segregating genetic variation”) What is this? this paragraph is very difficult to read, and I am not sure what is being done.

We have removed this paragraph.

Line 69: (Highlighted “both genetic variation”) Within? Again it is a negative statement which is more appropriate for the discussion.

We have removed this paragraph.

Line 74: (Highlighted “extensive phenotypic variation”) Why would this matter?

We have removed this statement.

Line 75: (Highlighted “morphotypes”) What is a morphotype?

A morphotype is a group of individuals with shared morphology — similar plumage traits in the case of Juncos. We now refer to these as Junco “taxa” rather than morphotypes to avoid confusion.

Line 76: (Highlighted “Fig. S1”) Does fig. s1 show this?

We have removed this reference to Figure S1 and, in fact, we no longer include this map at all.

Line 77: (Highlighted “This diversity....North America.”) This is one scenario but an alternative is that massive introgression has homogenized the genome except at key functional genes.

We have removed this statement.

Line 80: (Highlighted “environmental factors have been shown to partition genetic variation”) What does this mean?

This was referring to a previous genotype-environment association analysis (Friis et al. 2018), but we have removed this statement.

Line 82: (Highlighted “suggesting that considerable phenotypic diversity persists in the face of high gene flow”) Why? Others argue that rapid post glacial expansion accounts for the low differentiation.

We have removed this statement.

Line 85: (Highlighted “also”) In addition to what?

We have removed this statement.

Line 85: (Highlighted “groups”) ? what is a group?

We have removed this statement and no longer refer to Junco “groups” at all.

Line 93: (Highlighted “hierarchical”) ? what is meant here

We have removed this word.

Line 98: (Highlighted “influenced”) I think comparing the field acclimatization and long term lab differences is of major importance and should be emphasized.

We have devoted more space in the Discussion to this connection.

Line 100: (Highlighted “upon natural variation”) It does? Aren’t you asking if it does?

We have removed this statement.

Line 102: (Highlighted “population genetic structure”) What do you mean by population genetic structure?

We mean the distribution of allele frequencies among subpopulations. We have related the population genetic variation to environmental variation. To be more precise, we have altered our language in a number of places throughout the manuscript to indicate allele frequencies.

Line 104-106: (Highlighted “We then characterized fine-scale, range-wide population genetic structure within the Junco genus to determine whether it is influenced by the same climatic indices”) Again this seems out of place. it is basically a method to control for genetic differences, and the natural link is between thermogenic capacity in situ and lab acclimation

We have elaborated on our reasoning for conducting this analysis (Lines 89-96).

Line 107: (Highlighted “environmental heterogeneity”) Do you mean annual thermal regime??

Yes, we have reworded this statement to make this clear (Line 97).

Lines 110-112: (Highlighted “We predicted that junco populations that experience greater seasonal temperature variation would exhibit higher thermogenic flexibility than those from more thermally stable regions”) And that this would persist.

Yes, we hope this is now more clearly stated in Prediction 2.

Line 121: (Highlighted Msum) What is Msum!!!!

We have now detailed the quantification of Msum in the Introduction (Line 81).

Line 128: (Highlighted Mb) What is this!!!

We had defined Mb upon its first use, which occurred in the line previous to that which the reviewer highlighted (formerly 127, now 119).

Line 129: (Highlighted “morphotype”) Define this too.

Again, we have removed the word morphotype from the manuscript to avoid confusion.

Line 145: (Highlighted “were”) Methods?

We have moved this statement to the Methods section.

Line 152: (Highlighted “comprised”) This is a well known result from the Mila group. Not clear what is being affed.

We have removed this statement.

Lines 156-159: (Highlighted “In particular, redundancy analysis (RDA) is a powerful multivariate tool for identifying even weak correlations between genetic and environmental data⁵³. We thus performed an RDA to quantify the population genetic variance that partitions with climatic indices while controlling for background genetic structure. ”) Methods?

We have moved this statement to the Methods section.

Lines 172-179: (5 separate comments)

We have removed this paragraph altogether.

Line 190: (Highlighted “partial Mantel”) What are you holding constant here?

We have clarified this statement to read “partial Mantel test conditioned on geographic distance: $r = -0.40$ ” (Lines 162-163).

Line 194: (Highlighted “ $p < 0.001$ ”) Sample sizes. Here site should be a random effect?

We now state the sample size in the first sentence of this paragraph. The sample size was consistent for all analyses using the acclimated individuals. We have not included site as a random effect because each population is from a single site, thus temperature range would be entirely overlapping with site.

Line 195: (Highlighted “pre-acclimation Msum and Mb”) How and why, isn’t subtracting one from the other doing this

Yes, it is. This was meant as justification for why we subtracted one from the other, but we see this caused confusion and have removed this statement.

Line 206: (Highlighted “Conclusions”) Make this an integration with the Discussion

We have integrated this section with the Discussion as suggested.

Line 212: (Highlighted “This pattern was also replicated in the laboratory”) Wasn’t the previous sentence also referring to laboratory?

No, the prior sentence was referring to the field data. We have added language to make this clearer (Lines 183-186).

Line 214: (Highlighted “population genetic variation”) Means? ‘neutral’ Variation within populations? Usually this would simply be a consequence of population size

Our analyses show that allelic frequencies correlated with variation in temperature heterogeneity. We have tried to be more precise about our wording in the text (Lines 186-188).

Line 219: (Highlighted “to”) See comments above. There is definitely a latitudinal gradient in tolerance, and in seasonality, when comparing species.

We have removed this paragraph entirely.

Line 236: (Highlighted “sometimes”) So?

We do not understand what the reviewer finds problematic here.

Line 266: (Highlighted “made within 48 h of capture”) How were birds maintained during the time

We have added these details to Lines 318-319.

Line 267: (Highlighted “though”) And. Better to give the fraction

We have added these details to Line 320-322.

Line 268: (Highlighted “light phase”) explain

We now state “during daylight hours” (Line 323).

Line 274: (Highlighted “starting at 0°C”) to what?

We have added this detail to the text (Line 329). Sliding cold exposure started at 0°C and, after a 20 min initial exposure, temperatures were reduced at a rate of approximately 3°C every 20 min until birds showed a decline in metabolic rate indicative of hypothermia. Generally, hypothermia occurred within 30-100 min for the sliding cold exposure measurements.

Line 282-283: (Highlighted “exposed to acclimation experiments³⁶, or immediately euthanized and deposited in museums”) Always helpful to give numbers

In this case, we don’t think the numbers are relevant, as the fate of the individuals does not matter for the analysis. We provide the fates in the supplemental materials only in case someone is looking for *Junco* tissues.

Line 305: (Highlighted “running”) Explain what the running average is here, why is not just the average

We have removed the word “running” and now refer to them as averages.

Line 305: (Highlighted “Tmin, Tmax ,Tdrange”) High multicollinearity here

Yes, for this reason we did not include more than one environmental term at a time in our analysis. We have added details to make this clear (Lines 363-364).

Line 306: (Highlighted “according to ref 75 using”) We shouldn’t have to look up the ref. what was the standardization procedure?

We now include a brief summary of this standardization method (Lines 364-366).

Lines 307-308: (Highlighted “We then used AIC values to evaluate differences in model fits among environmental variables and with that of a null model”) But what is the regression model?

In addition to the verbal description of the model, we now include the equation for the regression model in the text (Line 368).

Line 308: (Highlighted “morph”) Remind us what ‘morph’ is

We have removed the word “morph” from the manuscript.

Line 309: (Highlighted “single ‘best’”) Why do you want to do this?

We did this because, as the reviewer points out above, these environmental variables are highly correlated, and we therefore chose to use only a single variable in the model at once.

Line 312: (Highlighted “effect”) I am not sure how it does it.

We have removed this statement and this analysis.

Lines 370-371: (Highlighted “In both RDAs, we tested for, but did not find, multicollinearity among predictor variables (variance inflation factor < 5 for all variables)”) But I thought that was what you did to select the predictors in the first place?

Yes, that is correct, but this additional step is commonly used in RDA to ensure that there is not collinearity in the model. In our case, the variance inflation factors were all quite low such that no additional terms were removed from the model, so it is rather moot. We simply report here that we checked for it.

Line 372: (Highlighted “ANOVA-like”) Means? Don’t you permute and for each permutation do something anova like?

This is the description that the authors of this method use to describe it. We have provided the reference in the text (Line 445, ref 77).

Line 374: (Highlighted “using variance partitioning”) Explain? You could use standard R-sq?

This method partitions the explanatory power of the different climatic variables used in the model by running multiple RDAs to determine the linear effect of each on the response data. In doing so, it calculated the coefficients of determination (R^2) for each partition, as well as that shared among partitions.

Line 396: (Highlighted “captured”) What about age?

Yes, thank you for pointing out this neglected detail. All individuals used in the acclimation study (and the field study) were adults. We have added this detail to both sections (Lines 307 & 459).

Line 435: (Highlighted “t-test”) Always give sample sizes

We have added the sample size (Lines 500-501).

Line 467: (Highlighted “We calculated pairwise genetic distance as $F_{ST}/(1 - F_{ST})$ ”) Why? This would be an isolation by distance model, but F_{ST} or D_{xy} would be the usual distance metric

Because we are performing a Mantel test to determine if genetic and environmental distances covary, we calculated genetic distance as $F_{ST}^* = F_{ST}/(1 - F_{ST})$ according to Roussett 1997 (as in an IBD model). This is standard practice when testing for relationships between genetic distance and environmental or geographic distances. F_{ST} itself is not a distance in the strict sense, but rather a measure of population differentiation. Importantly, the conclusion from this test (that genetic and environmental distances do not covary) does not change if we instead employ F_{ST} . This was only for the partial Mantel test; F_{ST} itself was used in the main analysis in which we tested variation in ΔM_{sum} using MCMCglmm.

Line 474: (Highlighted “phenotypic differences”) Reword...

We do not understand what the reviewer finds problematic here.

Line 479: (Highlighted “that allow for Bayesian”) Isn't it a Bayesian approach.

We have removed this clause.

Line 482: (Highlighted “pairwise F_{ST} ”) What is this? very unclear

We had defined pairwise F_{ST} prior to this in the text (now Lines 527-528). To make clear how we used the pairwise F_{ST} estimates in the MCMCglmm model, we have added the statement: “Because individual measures were used in the model, the pairwise F_{ST} among two individuals corresponded to that of their respective subspecies” (Lines 542-543).

Line 757: (Highlighted “background structure”) Is this term used elsewhere? The r-square is miniscule. Perhaps this is reasonable but need to explain

We now state that the model is conditioned on PC1 and PC2 in the table legend (Line 859). We discuss the seemingly small proportion of variation explained by this model in the Discussion (Lines 222-230). In brief, however, this is not an unexpectedly small proportion of variation to explain given what other, similar, studies have found and the fact that population structure is weak across *Juncos*.

Line 764-765: (Figure 2 Legend) Common names here, scientific elsewhere. Are these morphotypes? What does controlling for differences mean? Table S3 you could convert to an anova table (e.g. using `aov(model)`) rather than linear model with different references.

We have removed the use of common names throughout the text. “Controlling for mass” refers to the use of body mass as a covariate in the model. We now include the formula for the model that was used to generate the graphs to make this apparent (Line 848).

Line 768: (Figure 3 Legend) Must state what it is conditioned on. Everything driven by a few outliers.

We now state that the RDA is conditioned on PC1 and PC2 (Figure 2 legend).

We do not understand who the outliers are that the reviewer is referring to. In the case of the PCA, these plots are not showing outliers but rather the population genetic differentiation among individuals in our dataset. These two panels clearly show that some populations (e.g., *J. vulcani* and *J. bairdi*) exhibit greater genetic differentiation from the other individuals. This is expected given prior phylogenetic/population genetic work in the *Junco* genus.

Line 772: (Figure 4) Typo: *J. h. palliatus*

This is not a typo: *J. p. palliatus* is correct. As we now state in the Intro (Lines 82-84), *J. phaeonotus* and *J. hyemalis* are not reciprocally monophyletic.

Reviewer #3 (Remarks to the Author):

This paper tests the hypothesis that plasticity in thermogenic abilities varies with environmental heterogeneity – more variable environments promoting increased plasticity. The authors address these questions with a three-pronged approach that is quite impressive in scope. First, by measuring metabolic responses to an administered cold stress in the field; second, by measuring genetic variation (RADseq) across populations/species in the genus *Junco*; third, by experimentally assessing thermal acclimation across population in a common garden experiment in captivity. This study is important and timely. Furthering our understanding of how species and populations diverge in thermal acclimation is the foundation needed for predicting species persistence into the future.

We thank you for taking the time to review our manuscript and provide feedback. We appreciate your support of the article and have worked to incorporate your comments regarding seasonal differences in the revision.

These are the main results:

1. Populations of Juncos vary in thermogenic flexibility, suggesting that local environments shape phenotypic plasticity.
2. Annual temperature range explains the most genetic variation of the environmental metrics measured.
3. In a common garden study, individuals from more variable environments were able to mount a more robust thermogenic responses as measured by an increase in M_{sum} (peak metabolic rate under cold exposure).

Despite these strengths, I have three major concerns, chiefly:

(a) Birds are sampled from breeding and wintering seasons. Seasonal differences are well documented in Juncos. This does not appear to be balanced within species/morphotypes or controlled for statistically.

The reviewer is correct that our field sampling scheme is biased towards breeding birds, such that three taxa are sampled in both seasons and two are sampled in the breeding season only (none are sampled in only the winter season). The reason for these differences is largely logistical: in winter, juncos are harder to capture, many of the field sites are not easy to access, nor are juncos present at all of them, and our academic schedules do not allow much time for winter field work.

Importantly, however, the amount of daily temperature range is similar across seasons; winter birds do not occupy one end of the spectrum and breeding birds the other. We now use different symbols in Figure 1 (formerly Fig. 2) to clearly illustrate the breeding and wintering individuals. We therefore do not think that this sampling discrepancy is driving the patterns that we observe here.

Indeed, our current understanding from the literature is that changes in avian thermogenic capacity (and in juncos specifically) are triggered in response to temperature changes and not photoperiod cues (Swanson et al. 2014 J. Exp. Biol.). Our data support this hypothesis in that temperature range explained more variation in M_{sum} in the field than did daylength (Table S2). Moreover, we did test whether the addition of a term for season in the model would improve its fit and found that it did not. This analysis is now presented in the Results (Lines 120-121).

The acclimation study, which did more to control prior conditions/physiological state, provides further support that seasonal differences are not the most likely explanatory variable for geographic and/or taxon variation in M_{sum}. Under this controlled captive experiment, all individuals were acclimated to the same photoperiod and temperature conditions for 8 weeks. Here we also found that capture season did not influence flexibility (Lines 558-560), in agreement with the conclusions of our field-based results.

We have also added 43 additional breeding individuals (data which were previously not quantified) to help balance the sampling across subspecies such that we now have an even number of *J. h. oregonus* and *J. h. mearnsi* individuals, though the other three taxa have much smaller sample sizes. That being said, to our knowledge this dataset represents the largest

compilation of avian Msum values from a single species to date. Thus, the power of the analysis comes from this large number of individuals and the large amount of environmental variation that we captured across the sampling points.

(b) It is unclear how the museum samples were handled when addressing how genetic variation relates to climate - What time period of weather were the museum data correlated with, since some samples date back to the 1980s? And how does that contrast with the climate analyses? I'm unclear on the degree of matching here – do the genetic samples and environmental data cover the same time period, or are they sampled generations apart?

The museum samples that we used to generate the population genetic data were collected from 1983-2017. We used climatic data from the WorldClim dataset, which was aggregated from 1970-2000, in the genotype-environment association analysis. The two periods therefore do not entirely overlap. However, the climate data are meant to represent longer-term averages rather than a specific instant in time that would correspond to the precise sampling event.

(c) The writing and structure of the paper was, at times, hard to follow for a reader not yet familiar with the project. This is partly driven by the journal's constraints on the order of presentation and compounded by jargon that is not explained and/or explained too late (Msum, acclimatization, in situ, that the focus on the project is cold tolerance, etc.).

In light of the reviewers' comments, we have wholly rewritten and restructured the paper. We hope that the reviewer now finds it easier to follow. We have removed jargon (see below) and endeavored to briefly summarize our Methods in the Introduction so that the Results are more straightforward.

I have elaborated on these points and other more minor issues below (by line number)

15: Somewhere in the abstract, please state that your study is focused on thermal challenges in a cold capacity. This was not evident until the final paragraph of the introduction.

As stated above in our comments to Reviewer 1, we have not quantified cold tolerance or “cold capacity.” We therefore think the description provided in the Abstract is accurate. We have, though, restructured the Introduction to place more emphasis on thermal acclimatization.

first two paragraphs of the introduction: The framing of the paper put a lot of emphasis on developmental plasticity, and this was not returned to in the discussion, suggesting its not a relevant framing. I recommend the authors better use this space to set up the proposed work. I agree with the authors' argument that much of past research on plasticity has focused on development, but there is quite a lot of work on adult or moment-to-moment plasticity, which could be incorporated here too. I especially encourage more citation of non-bird or even non-animal work.

We have removed these two paragraphs from the Introduction, as well as all discussion of developmental plasticity.

52: “Unlike developmental plasticity, phenotypic flexibility...” Are the authors trying to differentiate different temporal scales of plasticity? Is phenotypic flexibility meant to signify something different from phenotypic plasticity?

We have removed this statement and all discussion of developmental plasticity.

61 & 69: These statements of novelty seem overstated, given past work in plants or ectothermic animals. I’m thinking for example of Lowry et al.’s (2019) PNAS paper on switchgrass across most of North America or Whitehead’s work on killifish.

We have removed the statement that was formerly Line 69. We have also combed the literature in search of additional empirical demonstrations to cite here but have not found much additional support for this hypothesis as it pertains to phenotypically flexible traits (i.e., acclimatization capacity). We agree that Whitehead’s work would be a logical source of such a citation—his group has certainly shown local adaptation in physiological traits across environmental gradients. However, they don’t usually quantify the magnitude of flexibility that populations exhibit. We went so far as to contact Andrew Whitehead himself to ask if we were overlooking any of his work, but he said we were not. He did, though, suggest that perhaps Griffith 1974 “Environment and Salinity Tolerance in the Genus *Fundulus*” (now ref 10) might be an appropriate reference. Griffith shows that *Fundulus* populations from brackish water show greater salinity tolerance, which could imply a greater magnitude of acclimatization ability (though this was not shown). We do not see the relevance of Lowry et al. 2019 in this context. This paper also provides evidence of local adaptation across an environmental gradient, as well as the genetic loci contributing to local adaptation, but unless we are mistaken they do not include measures of phenotypic flexibility.

62: Please clarify how this study accounts for “non-independence among populations (due to shared common ancestry and ongoing gene flow)”. I assume the authors are simply advocating for appropriate phylogenetic methods, but please clarify, as some readers might also think that, say, spatially explicit statistical models would account for this as well.

We have moved this statement to the Discussion and have added “by incorporating measures of population genetic differentiations in our acclimation analysis using Markov chain Monte Carlo generalized linear models” (Lines 239-242).

82: To what degree does the lack of genetic variation in *Junco* subspecies undermine the premise of this work? For a generalist audience, it would help to understand the scope of genetic variation seen here, and how it compares to other work, though this becomes more relevant later in the manuscript for those not already familiar with the previous studies on population genetics in *Junco* from the Mila research group.

The relative amount of genetic variation should not be relevant, as our models only seek to explain what variation does exist. Thus, the question is, given the amount of variation that exists among *Junco* populations, how much of it is explained by our environmental variables. Furthermore, as we make clear in the Discussion, the amount of variation that we do explain

using our environmental variables is entirely in line with other studies using similar analyses. We have, however, deemphasized the previous findings from the Mila group, as we think that they only served to confuse the reader.

90: For the generalist reader, I recommend defining Msum more clearly by stating what it stands for: summit metabolism.

We have made the recommended change (Lines 81-82).

91-3: Is “thermogenic capacity” synonymous with thermogenic performance? Please clarify. This comes up elsewhere in the paper too, related to thermogenic flexibility.

We have removed the term “thermogenic performance” from the manuscript to avoid confusion.

104: “in situ” is a broadly used term in biology, and it means completely different things depending on the discipline. I understand that “in the field” is not exactly accurate (or is it? I’m unclear on how birds were housed when pulled from the field in the first study). However, this made it hard to understand the results and discussion without also reading the methods, which come much later. Please clarify the language, so the reader can understand key points as they go.

We have removed “*in situ*” and now use “in the field” throughout the manuscript.

107: “Laboratory acclimation” is also unclear. The degree of jargon or otherwise undefined terms in this final paragraph of the introduction made it hard for me to know exactly what was done here, until I also went and read the methods.

We have changed this to “an acclimation experiment in the lab” (Line 96).

section beginning 118: I appreciate that everything cannot be included in the main text, especially with word limits and an integrative 3-part study like this... but knowing generally that this occurred in certain seasons is important. Please include that in this paragraph.

We have added more details to this sentence to make this clear (Line 115), as well as the analysis testing the model fit with the season term (results shown in Lines 120-121) and the use of different symbols to note breeding/non-breeding in Figure 1.

131: Result is hard to interpret without some sort of even balanced sampling across seasons, even within a morphotype. I also had to spend a lot of time reading through SI data tables to match species/subspecies with seasonality and location. I am concerned that there appears to be imbalance in the seasonal sampling across populations. How can we compare species or morphotypes if some are sampled only in winter and other only in summer?

We have addressed this comment above in the reviewer’s general comments. Again, no taxa were sampled in only the non-breeding season; all taxa were sampled in the breeding season. We hope that the changes that we have made to Figure 1, including the use of the open and closed

circles denoting sampling season and the addition of the map of sampling sites, helps to illustrate which taxa were sampled where and when.

131: Switching between common and scientific names is disorienting. Please choose one to use consistently throughout the figures and tables and corresponding text.

We apologize for any confusion this created. We have removed all common names and now use scientific names only. We can easily assign individuals to subspecies if they were sampled on the breeding grounds or if they exhibit plumage characteristics typical of a single subspecies. This is true for all individuals in the acclimation study. However, in the case of a few individuals sampled on the nonbreeding grounds in the field dataset, the subspecies is uncertain. This is because multiple subspecies that are lumped into the *J. h. oregonus* phenotype (known to birders as “Oregon Juncos”) have similar plumage characteristics as well as overlapping distributions in the non-breeding season. There are only a few of these individuals and they may be the same subspecies as the other *J. h. oregonus* group members that we sampled on the breeding grounds (*J. h. montanus*). We have therefore grouped all *J. h. oregonus* together and call them the *J. h. oregonus* group for accuracy. We have removed all other mentions to ‘morphotypes’ throughout the manuscript.

137-138: This is a compelling result, but I am still worried about the differences of seasonality in the birds were sampled.

Season was not a significant term in the model, as stated above.

Figure S1: I would really appreciate your Figure S1 map of the sub-species ranges included in the main text if there is room. Or perhaps there is some way to convey this alongside the existing map? A broad audience may not be able to retain the geographical areas of each species/subspecies and therefore may not be able to interpret your results without frequently flipping pages or referring to your supplemental. Consistent terminology may help to resolve some of this.

We felt that the figure formerly included as S1 was very unsightly and therefore chose not to include it in the main text. Instead we have taken the reviewer’s advice and each figure conveying the three types of data now includes a map of associated sampling sites. We also use consistent terminology (subspecies names) throughout the manuscript.

157: Clarify goal of the RDA. Why exactly is this needed beyond the PCA? And, how meaningful is it for each dimension to explain a maximum of 0.6% of the variation (R^2 , conditioned = 0.0060 for temperature range).

The PCA does not include any environmental data. The goal of the RDA is to identify associations between the genetic dataset and environmental data. This amount of variation is within the realm of expectation for an analysis of this kind. We have expanded upon this in the Discussion (Lines 222-231).

161: “conditioned” and “unconditioned” are not completely clear yet. Because of the order of

this paper/this journal, you need to walk the reader through your terminology before your methods.

We have removed the unconditioned RDA from the manuscript, as it seemed to be creating confusion and the results were qualitatively similar between the two. We therefore no longer need to refer to the “conditioned” model as such. We now walk the reader through this model before presenting the results from it (Lines 144-149).

187-188: Please clarify this sampling regime. I believe the authors purposefully selected non-migratory animals for this third study, to make sure they could know their temperature history. This seems like a biased sample within the Junco. Genetic variation (that stems from temperature variability) must be different in populations that experience extreme climatic variability. If the authors can show that the non-migrants have unbiased variability, then this study design would seem more appropriate. Ultimately, this tempers my enthusiasm for linking the genetic and trait-based portions of this paper, because the climate regimes shaping a migrant sampled in Canada (who winters in the US somewhere) may be totally different from the climatic regimes shaping a bird who spends in whole life just in the Black Hills.

Yes, we did purposefully sample non-migratory animals in order to reliably reconstruct the temperature variation that they experienced throughout the year. We do not expect that the relationship between flexibility and environmental heterogeneity differs between migratory and non-migratory birds. Rather, the reason to avoid migratory birds for this analysis is that we do not have information about the annual thermal environment they experience due to uncertainty in their movements at the spatial scales at which we can extract environmental data. While we do know that many subspecies move south in winter, we cannot say for sure where the birds that bred near Teton National Park spent their non-breeding season. Using populations that were likely non-migratory greatly simplified this issue.

We agree that it is reasonable to assume that some non-migratory populations (like those of the Black Hills) may experience more climatic variability than some migratory populations. Within our non-migratory populations, however, we also see variation in the annual temperature range experienced, such that some non-migratory populations experience more climatic variability than others. It is therefore unclear to us why the reviewer thinks this is a biased sample for our purposes.

In terms of the genetic data, we cannot know the extent to which individuals in the genetic dataset are or are not migratory. However, we are confident that we have included both migratory tendencies. While the *J. hyemalis* individuals likely represent variation from non-migratory (e.g., two San Diego individuals), to altitudinal migrants, to longer-distance migrants, many of the other taxa exhibit little to no migratory tendency. Yet these non-migratory individuals are distributed across orthogonal space in Figure 2d. We therefore do not appear to be targeting differences between migratory and non-migratory birds with this analysis, as the reviewer may be suggesting.

We acknowledge in the manuscript that there are shortcomings in the genetic dataset resulting from this uncertainty in annual thermal regimes. With these caveats in place, we do not think that we have overstated the conclusions that we can draw from these genetic results.

216-7: I'm not convinced we can conclude environmental heterogeneity is an important selective force. I need to know more to be able to assess this claim. How much variation are we explaining, and is this an important 'amount'? What is the scope of these R²? Is the # of SNPs high or low? Is this a real but small effect? Please contextualize with other work.

As we now state in the Discussion (Lines 222-225), "Together, the eight climatic variables explained a small amount of total genetic variation. However, this amount is well within the range of variation expected from a genotype-environment association analysis (e.g. refs. ^{34,35,42}) given that we only surveyed 2% of the genome with our SNP dataset." On top of this reduced sampling of the genome, we may only expect a small proportion of the genome to have experienced selection at an intensity that is detectable by this approach. That is, there are certainly some loci that are strongly structured by environmental variation, but they are washed out at this scale by a lack of structure in the genomic background across the group.

219-221: It may be true that these correlative studies have not been done in ectotherms, but there are studies that have looked at plasticity differences in ectotherms and plants resulting from environmental differences. The novelty of this study seems overstated in light of this.

We have removed this paragraph from the manuscript.

220, 223, 226, 230, 232, etc: Please choose either "plasticity" or "flexibility."

We have removed references to plasticity throughout.

257: Explain why birds from wintering and breeding grounds can be comparable. I have concerns with comparing birds of different physiological states without controlling for seasonality.

Both the breeding and nonbreeding grounds presented similar amounts of temperature range. Additionally, we have controlled for seasonality by including season as a term in the model (see above comments) but this term was not shown to improve model fit, implying that temperature range (and perhaps body mass) account for the variation in Msum exhibited among seasons.

267: Did you test for an effect of sampling time (within 48h vs within 24h)? This seems like a wide time frame knowing how much stress hormones should interact with metabolism, particularly for a species that has been widely studied for its variation in corticosteroid reactivity (which can vary among populations and seasonally). Please clarify so that the reader has more confidence in there being stable differences among populations in their peak metabolic capacity.

We are not aware of any study that has quantified the effect of short-term captivity on Msum over a time scale shorter than two weeks (Swanson and King 2013 *Curr. Zool.* 59). However, there is no effect of duration of captivity on Msum in our dataset. This is true if we include

captive duration as a linear or a categorical variable (0, 1, or 2 days). Only 15 individuals were held for 2 days and these were spread across three different sites; we now include this detail, as well as the mean amount of time individuals were held, in the Methods (Lines 321-322).

274: “Static cold exposure:” I assume this means that you exposed birds to cold suddenly. Did the authors present how the birds were housed (what temp) during the 24-48 after capture?

The “static” term refers to birds exposed to a single, unchanging, temperature in the heliox atmosphere, to distinguish it from a sliding cold exposure where birds were exposed to a declining series of temperatures. These are the same terms used by the Swanson et al. (1996) paper initially defining these strategies for cold exposure. In the present study the birds were all held indoors (Line 319) but the field station conditions varied across sites, such that we were not able to control for temperature during this period. Again, we did not find an effect of captivity duration on these measurements.

288: Define “acclimatization” or use a simpler term like “recent thermal experience” or “recent thermal history”.

We respectfully disagree with the reviewer about changing this term. Acclimatization is a widely used word in biology: a Web of Science search (dated 27 March 2021) reveals 8,981 references to “acclimatization.”

300: Please explain more about the climatic windows. 7 to 14 days is quite a wide range. Were some windows days 0 to 7 prior to capture and others were 0-14 days prior to capture? Or did you measure days 7-14 prior to capture? In this case, why were the most recent 7 days prior to capture thrown out? I was unclear on exactly how this was done from the main text.

These windows ranged from 1-7 to 1-14 days prior to capture, thus we tested 7 different temporal windows. We have added language to clarify the time windows used (Lines 355-357). Originally, we had included the day of capture (day 0) as the start date in the analysis, but upon reflection this didn't seem appropriate since many individuals were caught early in the morning and thus the day's weather likely had little effect on their current acclimatization. Importantly, the results are the same using either the day of capture or the day before capture, but now we only present the latter.

349: Please explain why sites that departed from Hardy-Weinberg equilibrium were filtered out. Also, minor typo: I believe it's 'berg', not 'burg'.

Yes, indeed. Thank you for catching this.

354: This topic sentence would be helpful on line 155. Consider moving.

We have added a similar topic sentence to the beginning of the genotype-environment association results (Line 138-140).

359: What was the time frame that the weather data was averaged across (how many months) for

the genotype-env association analyses? I.e., 12 months before each muscle sample was collected? How was this managed since the museum tissues were sampled across decades?

As detailed above, the climatic data that we used are from the WorldClim dataset, which includes interpolated climate averages over the period from 1970-2000. Thus, they were averaged over years rather than months, and the genetic and environmental data are not paired directly in time.

Related to this point, is historical time balanced by lat-long in the sampling regime? Is season balanced within in the sampling regime? I understand that RADseq is independent of when an animal is captured, but this does matter when we decide what climate to assign to that bird. This is particularly important considering that some museum samples are from a very long time ago (decades), before some of these populations even bred in their current locality.

Historical time was not balanced across lat-longs. Our aim was to maximize the geographic extent of the sampling and we therefore did not limit our samples by collection date. However, we only used samples for which tissues were available, which means no samples were collected before 1980.

We also aimed to include tissue samples from the breeding grounds only. Most samples were collected May through early August, including all samples collected in the U.S. and Canada. The samples available from Mexico were much more limited and thus some of the individuals included fall outside of this May-Aug range ($n = 11$). However, these populations are not known to migrate (Sullivan 2020 *Birds of the World: Yellow-eyed Junco*) and thus we do not have reason to believe that they were not on the breeding grounds.

It is not clear to us what the reviewer means by “before some of these populations even bred in their current locality.” We have included two individuals from the city of San Diego, which are part of a population that colonized UCSD’s campus in the recent past (1980s): perhaps this is what the reviewer is referring to? The two samples we included were collected in 2004. Again, we believe that all individuals were sampled on the breeding grounds. The dates that we have reported in Table S3 correspond to the those provided by the museums.

362-366: Consider renaming environmental variable to more intuitive abbreviations.

These are the conventional names provided in the WorldClim dataset. We only name them in the Methods for those familiar with the dataset, but do not refer to these abbreviations elsewhere.

366: How do these environmental metrics relate to migratory status? Please clarify: are you using climate data from the place where the bird was captured?

Yes, we are using climatic data that corresponds to the site of capture for each tissue sample (Lines 432-434). We do not know whether these individuals are migratory or not, but again, we selected samples from the breeding grounds. We cannot know where these individuals spent the rest of their year and we address this shortcoming with the dataset in the Discussion (Lines 228-231).

We also repeated our genotype-environment association analysis using only climatic data from the breeding months (i.e., May through July for consistency with when most of the samples were collected) to try to address this concern. We again retrieved this data from the WorldClim dataset, which includes monthly interpolated climate data aggregated over the same time period (1970-2000). In this case, the monthly climatic variables available are different than those available in the annual dataset that we used previously. They are minimum temperature, maximum temperature, mean temperature, precipitation, vapor pressure, wind, and shortwave radiation. We repeated the redundancy analysis using these variables to explain variation across our SNP dataset and found that they explain less than half of the variation explained by the annual variables. We therefore think that, although summarizing the climate across the year at the site of capture it is an imperfect solution, it does encompass more of the climatic variation that individuals likely experienced than restricting the analysis to the breeding season alone. For this reason, we do not present this secondary analysis in the text.

368: Does “partial RDA conditioned on background population structure” mean it accounts for genetic differences and phylogeny? Using something external or this specific dataset?

The partial RDA was conditioned on the first two axes of the principal component analysis that we performed. We have tried to make this clearer in both the Results (Lines 144-146) and the Methods (Lines 442-443).

403: "Erase" is an overstatement. Please soften this term.

We have changed this to “minimize” (Line 475).

404-6: Please elaborate how this result shows that birds are not making seasonal adjustments. If Delta AIC is under two, then we cannot reject this as a parsimonious model, though I recognize that this p value suggests this variable is not meaningful here.

As Arnold 2010 (J. Wildlife Management) lays out, one should not include uninformative parameters in models that otherwise do not vary, even if their AIC scores are similar. So, here, the top model includes all of the sample predictor variables as the second-best model with the exception of season. Since season is not a relevant predictor variable (i.e., its p-value is >0.05 and its 95% confidence interval crosses 0 [Line 121]), and it is only included in the second-best model, there is no reason to interpret its effects.

415-416: Do the species differ in whether they were in breeding condition, suddenly experiencing a lengthening, or suddenly experiencing a shortening in day length based on the switch from capture site to 12:12 in the lab? I cannot help but think about how much is changing within the birds' physiology during that adjustment. How was season treated in the aforementioned model in my last comment?

The four populations that were captured during the breeding season experienced a shortened daylength (by ~2-3 hrs) upon introduction to the lab, while *J. h. aikeni*, captured during the non-breeding season, experienced a near constant daylength.

We have now tested whether capture season influenced pre-acclimation M_{sum} and found that it did not (Lines 551-554). This provides strong evidence that our eight-week adjustment period did indeed appropriately reduce differences among individuals. Similarly, we found that capture season does not influence ΔM_{sum} (Lines 558-560).

422: Does the regression of brood patches and cloacal protuberances indicate that after the adjustment period there were no biological differences between breeding and wintering individuals? The timing here is unclear and incredibly important. Are these birds regressing, recently regression, or just prior to recrudescence?

Again, our intention was that the 8-week adjustment period reduced differences between breeding and non-breeding birds. We have removed any reference to brood patches, cloacal protuberances, or gonadal regression because we understand the reviewer's concern that these are not definitive indices. The most important point to convey, and which we now present, is that our analyses show that breeding status at capture did not influence M_{sum} before acclimation (Lines 551-554) or flexibility in M_{sum} (Lines 558-560).

Furthermore, even if potential differences in the reproductive status among populations had persisted for eight weeks under common conditions, we have controlled for these potential differences in our experimental design: We employed control treatments for each population, as well as measurements before and after acclimation for all individuals. Additionally, we replicated this across five populations. The model therefore accounts for potential differences in the starting point of each individual (by using the difference observed over the acclimation period) and is focused on the interaction between treatment and population.

436: I am not convinced we can reject the idea that birds improved in their cold tolerance (or vice versa) because the challenge changed.

First, we did not quantify cold tolerance and do not claim to show differences in cold tolerance here (addressed also in comment to Reviewer 1). Second, while the cold challenge did change, we did not find variation in M_{sum} using these different temperatures across the Control treatment. Control birds did not exhibit heightened M_{sum} when exposed to the lower temperature, which indicates that the two procedures provided similar levels of cold challenge for quantifying M_{sum} (Lines 497-501).

Figure 1: color dots are hard to parse out, consider enlarging the figure of adjusting the format for more clarity so that multi-colored dots are not obscured.

We have removed this figure and now show each sampling scheme separately in Figures 1-3.

Figure 2: I prefer figures captions to not require reading the text to understand. Please define M_{sum} and M_b . Please include the seasonality (breeding/winter) and sampling location in this figure for each of the subspecies/species. If I got it right from looking through the SI material, yellow-eyeded are only sampled in spring, and gray-headed are only sampled in non-breeding.

Please also restrict the trend line to extend only through the range of data.

We have expanded the figure captions to include these details. We do not wish to restrict the trend line because we find that when we do so, the line is obscured by the data points. In addition, the trend lines results from a model in which all five taxa were included, such that the range of temperature variation is from 9 to 23 degrees. This encompasses most of range for which the trend line is shown.

Figure 3: I had to reference supplemental to understand this figure. I'm not convinced that the PCA plots (a and b) add much, except to reiterate that there really isn't much genetic differentiation among most samples.

We have included the PCA plots both to show that *J. hyemalis* and *J. phaeonotus* group together in our dataset and to show which axes were used to condition the RDA. We have expanded the figure legend and now include a map of the sampling points with a consistent color scheme in a separate panel.

Figure 4: This is a very cool result! Please include location and seasonality of each population. *J. h. aikeni* were the only birds sampled in March (all others in July). Could these birds be already primed to respond to cold? But the general result still holds even with omitting the *J. h. aikeni* data, so I suspect this cannot explain the pattern.

We have expanded this figure to include a map of the sampling locations. Again, we tried to minimize the prior acclimatization of each population by holding them for eight weeks before beginning the treatments and we now show that capture season did not influence flexibility in Msum.

Supplemental: Why does Table S3 present 5 different models with different reference-morphotypes? Please clarify.

We have removed this table.

REVIEWERS' COMMENTS

Reviewer #1 (Remarks to the Author):

I am satisfied with the authors explanations, justifications, and changes in response to my previous questions. I am particularly excited about the new analysis and result added with S1 and that it mostly supports the previous findings!

Reviewer #3 (Remarks to the Author):

I commend the authors on a well-reasoned response and thorough revision. Although there were a few small points where we disagreed, the newly added data, analyses, and citations assuage my prior concerns. The substantially revised introduction and discussion make for a beautiful piece of scientific prose that highlights the importance of the work, while acknowledging limitations as well.